# MIMIC: a flexible pipeline to register and summarize IMC-MSI experiments
Reto Gerber [1,4], Jake Griner [2,4], Silvia Guglietta [2], Carsten Krieg [3] ✉ & Mark D. Robinson [1] ✉

Spatial omics is transforming our ability to interrogate local tissue microenvironments by enabling spatially resolved measurement of biomolecules such as transcripts, proteins, and metabolites. However, capturing the full biological complexity of tissues often requires combining multiple modalities, which introduces both experimental as well as computational challenges. To address computational difficulties due to differences in resolution, noise levels, and available channels, we present MIMIC: a reproducible, semi-automated workflow that integrates Mass Spectrometry Imaging and Imaging Mass Cytometry for joint downstream analysis. MIMIC incorporates rigorous quality control, including registration error assessment, and supports pixel-level modeling to delineate analyte-cell type associations. We demonstrate the power of our approach with a proof-of-concept study on artificial tissue and apply it to human liver tissue affected by metabolic dysfunction-associated steatotic liver disease. Despite integration challenges, MIMIC provides a robust framework that successfully recovers known molecular associations and reveals novel spatial relationships across modalities.

The understanding of cellular niches has long been central to studying tissue formation, regeneration, and disease initiation. Traditionally, pathologists and biologists relied on morphological features to explore tissue structures, later advancing into immunohistology to focus on specific proteins or cell types of interest. However, the rise of advanced proteomics, transcriptomics and innovative multiplexed tissue-staining technologies is breaking new ground in tissue characterization by allowing the simultaneous measurements of different analytes ultimately contributing to a more comprehensive understanding of tissue heterogeneity.

To push the boundaries of tissue analysis, spatial biology techniques now allow for a comprehensive interrogation of tissue heterogeneity across scales and modalities. The integration of multiple modalities provides a multidimensional view of tissue biology with a more complete depiction than traditional approaches, thus leading to a deeper understanding in various application areas, such as development, disease initiation and progression, diagnostics, therapeutic strategies, and precision medicine. Emerging high-throughput methods for molecular analyte detection are paving new paths[1]. Among them, spatial transcriptomics can reveal RNA-level changes, while highly-multiplexed antibody-based platforms offer insights into protein-level modifications[2]. In addition to these are mass spectrometry imaging (MSI) techniques, which capture spatio-temporal changes in glycans, lipids, metabolites, proteins, and even drugs[3].

There are several emerging MSI techniques, such as matrix-assisted laser desorption/ionization (MALDI)[4] and secondary ion mass spectrometry (SIMS)[5]. While MALDI-MSI is a soft ionization technique allowing the analysis of large intact molecules at a resolution between 5 and 100 $\mu m$, SIMS-MSI is a hard ionization technique that fragments large molecules but achieves a higher resolution (0.05–2 $\mu m$)[6]. Despite providing spatio-temporal details and unbiased information on tissue-specific glycans, lipids, metabolites, MSI techniques do not allow the identification of the specific cell types. Conversely, antibody-based highly multiplexed imaging techniques such as CODEX, MIBI, PhenoCyler, COMET, GeoMx, IBEX, Orion, and imaging mass cytometry (IMC) allow the identification of virtually all cell types with high precision. The combination of cell position and cell type label facilitates various downstream analyses such as spatial signature analysis, pairwise cell interaction analysis and cellular microenvironment analysis[7]. Despite their numerous advantages of these high dimensional imaging techniques, when thinking along the '4S-criteria for performance' (speed, specificity, spatial resolution, and sensitivity), there are certain trade-offs that are important to consider[8]. For MSI data, higher spatial resolution generally comes at the cost of lower sensitivity, thus potentially missing lower abundant analytes. For multiplexed imaging, measuring more markers usually requires longer acquisition times (e.g., sequential immunofluorescence) or can result in lower spatial resolution (e.g., IMC). Furthermore, in the case of spectrometry-based imaging

[1]Department of Molecular Life Sciences and SIB Swiss Institute of Bioinformatics, University of Zurich, Zurich, Switzerland. [2]Department of Regenerative Medicine and Cell Biology, Medical University of South Carolina, Charleston, SC, USA. [3]Department of Pathology and Laboratory Medicine, Medical University of South Carolina, Charleston, SC, USA. [4]These authors contributed equally: Reto Gerber, Jake Griner. ✉e-mail: kriegc@musc.edu; mark.robinson@mls.uzh.ch

technologies that require laser ablation (e.g., IMC), partial tissue loss occurs. However, these high-dimensional tag-based imaging techniques provide no information about difficult-to-tag molecules such as lipids, glycans or metabolites. Therefore, the combination of MSI and highly multiplexed techniques is emerging as a powerful tool for a comprehensive characterization of tissues, spanning cellular and molecular composition. However, to achieve effective and meaningful results from the combination of these techniques, the experimental setup needs careful planning. Most importantly is the consideration of whether multiple modalities can be acquired on the same tissue slice or not. Measurements on the same slice are potentially less biased and easier to integrate, but are more difficult to achieve experimentally. Measurements on adjacent slices could lead to biases (e.g., 3-dimensional structures), but the advantage is a usually simpler experimental setup with reduced optimization steps. Furthermore, some experimental setups may *require* adjacent sections, such as MALDI imaging of lipids in fresh frozen tissue combined with IMC; it was shown that an initial MALDI-MSI may damage the sample and prevent antigen recognition, while IMC first involves the use of detergents that will greatly reduce the lipid signal[9].

Several variations of combining MSI with another modality on the same tissue section are now available, including MALDI-MSI plus histopathology and wide field autofluorescence[10], MALDI-MSI plus light microscopy[11], MALDI-MSI plus IMC[12], MALDI-MSI plus CODEX[13], MALDI-MSI plus multiplexed immunofluorescence (IF) microscopy[14], MALDI-MSI plus multiplex immunohistochemistry (IHC)[15], MALDI-MSI plus MALDI-IHC[16], MALDI-MSI plus Xenium[17], or TOF-SIMS plus IMC[18]. Methods that work with consecutive tissue sections include MSI plus Stereo-seq and targeted sequential IF[19], MALDI-MSI plus multispectral imaging[20], or MALDI-MSI plus spatial transcriptomics and IMC[21].

From a computational perspective, two possible integration (co-registration) paths to combine MSI pixels with another modality are prominent. First, ablation marks can be detected on 'PostMSI' microscopy images (i.e., after MSI acquisition), followed by matching to the MSI pixel grid[10,11]. Ablation mark detection can be done manually[10] or within an automated workflow[11]. Second, direct registration of MSI to another modality can be done, but it is challenging if the spatial resolutions differ and no corresponding channels are present[12,18]. The use of TOF-SIMS instead of MALDI allows both limitations to be overcome due to the higher resolution and the inclusion of a measurable DNA channel[18], while in other cases, the same (tissue) structures must be observed in both modalities[12,22,23]. Similarly, registration of MSI to H&E can be based on manually selected landmarks[20].

Several challenges remain with existing computational workflows (Supplementary Table S1). For example, `regToolboxMSRC` does not allow an IMC-to-microscopy registration, since it depends on intensities for co-registration, and no analogous channels are available[10]. The same holds for `SpaceM`, developed for single-cell metabolomics on cell cultures with MALDI-MSI plus light microscopy on the same slide, and therefore does not include capabilities to co-register adjacent slices, but on the other hand has automated MSI ablation detection, which could also be applicable on tissue sections[11]. `scSpaMet` does not include MSI ablation mark based registration and relies on the same resolution and analogous channels for MSI and IMC, which makes it currently incompatible with MALDI-MSI[18]. Direct manual registration of MALDI-MSI pixels to IMC can be used although it remains unclear what co-registration precision can be achieved and how larger MSI pixel sizes influence this precision[12]. `msiFlow`[22], on the other hand, automatically co-registers IF images to a lower dimensional projection of MALDI-MSI images of lipids by using high resolution (2 $\mu$m) transmission mode MALDI (t-MALDI2), a technically challenging modality to acquire that is limited to very specialized lab settings. Registration between MSI and H&E has relied on manually-selected landmarks, but we cannot expect a high precision due to the low MSI resolution[20]. Interactive, partially-manual registration has also been used, but rigorous quality control metrics for the precision of registration were not provided[19]. Therefore, solving the "pixel gap" when combining MALDI-MSI (where reaching single cell resolution is currently challenging) with single cell based

technologies remains a challenge. After successful co-registration, downstream analysis can involve among others statistical testing of associations, lower-dimensional projection via factor analysis[24], or prediction of single-cell omics[25].

In this work, we present a reproducible (semi-)automated workflow called MIMIC, Mass Imaging Modality Integration Co-registration, for the joint analysis of mass imaging modalities, combining MALDI-MSI with IMC using a series of before/after microscopy images as a scaffold. It overcomes current limitations by providing precise automated co-registration together with stringent registration quality evaluation. Furthermore, we provide statistical models to associate cell types and MSI-measured compounds at a slide- or domain-level, the result of which can be introduced into models for cross-condition discovery. The workflow has been developed with experimental setups where the two modalities are acquired either on the same or consecutive tissue slices. Both automated as well as manual registration between both MSI-to-PostMSI and IMC-to-PostIMC are supported, which allows for greater flexibility.

## Results

### A flexible pipeline to register MSI-IMC experiments

A typical MSI-IMC experimental workflow is shown in Fig. 1a, where microscopy slide scans are acquired (denoted as PreMSI and PreIMC) separately for each of two adjacent tissue slices. MSI and IMC measurements, via ablation of a selected region, are independently taken, and microscopy slide scans are acquired again (denoted PostMSI and PostIMC). One potential simplification is when a single slice can be used for sequential measurement of MSI and IMC (dashed line in Fig. 1a), whereby PreMSI and PreIMC represent the same image.

An overview of the computational pipeline from 4 (or 3) microscopy images as well as the MSI and IMC acquisitions, to tabular and statistical summaries is shown in Fig. 1b (and Supplementary Fig. S1). Raw MSI spectra are processed into multi-channel images, including normalization, peak picking, binning and potentially log transformation (Box A in Fig. 1b, see Methods). IMC data are segmented using an existing pipeline[26] to obtain cell masks and annotations, and includes both pixel-level and cell-level representations (Boxes B1 and B2 in Fig. 1b, see Methods).

The main challenge revolves around *registering* the MSI and IMC modalities via the chain of images. The standard approach is to treat PostMSI as the canonical reference, and via a set of co-registrations, link the quantitative MSI to IMC modalities (i.e., as shown in Fig. 1b: MSI $\rightarrow^{C1}$ PostMSI $\leftarrow^{C2}$ PreMSI $\leftarrow^{C2}$ PreIMC $\leftarrow^{C2}$ PostIMC $\leftarrow^{C3}$ IMC). Registration of MSI-to-PostMSI consists of detecting ablation mark centroids on the PostMSI image and registering them to the MSI pixels (Box C1 in Fig. 1b and Supplementary Fig. S2; see also Methods). Manual selection of corresponding landmarks is supported. Co-registration of the microscopy images is achieved in multiple steps (Box C2 in Fig. 1b). First, a PostIMC-to-PreIMC registration is done using an affine transformation. Second, the PreIMC-to-PreMSI step uses first an affine transformation and then a b-spline transformation. Lastly, the PreMSI-to-PostMSI registration is conducted using an affine transformation. If a single slice is used, no registration between PreIMC and PreMSI image is required. Registration of IMC to the PostIMC microscopy image is done by automatic or manual detection of the IMC ablation area (Box C3 in Fig. 1b; see also Methods). All individual steps in the workflow are executable via Snakemake[27], with the final processed data being quality control summaries and the integrated pixel-level data that can be used for downstream analysis.

### Experiments to evaluate registration performance

Since the results of downstream analysis tasks are expected to depend heavily on the registration, it is important to comprehensively evaluate registration quality. We present a case study consisting of MALDI-MSI of glycans at 20 or 30 $\mu$m resolution on liver explants from two tissue microarrays (TMAs, see below). In particular, the distances between matching landmarks (See Methods, Supplementary Table S2) for the five co-registration steps of MIMIC are shown in Fig. 2a (DICE coefficients are

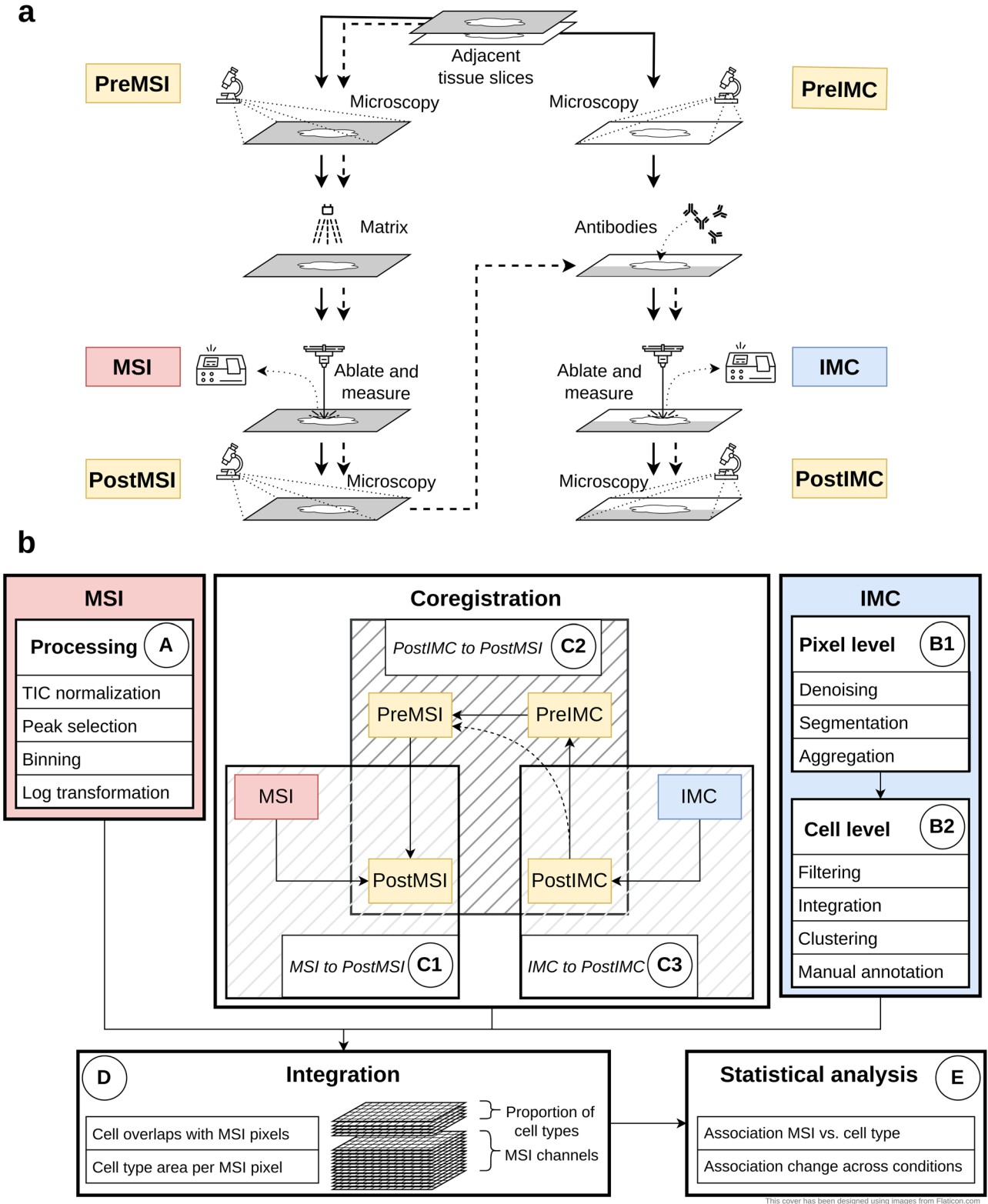

**Fig. 1 | Overview of the analysis workflow. a** Experimental workflow. Starting from the top with adjacent tissue slices the workflow follows the solid lines, acquiring the different modalities in parallel. For a single tissue slice, the workflow continues after postMSI acquisition with IMC measurement (dashed lines). **b** Computational workflow consisting of five main blocks: analysis of MSI data (A), analysis of IMC data (B1-2), Co-registration (C1-3), Integration (D) and combined analysis (E1-2).

given in Supplementary Fig. S3). The median landmark distance (MLD, see Methods) between the MSI pixel coordinates and the corresponding abla-tion mark centroids (Fig. 2a MSI-to-PostMSI) is around 2.5 $\mu$m. The result is similar for the IMC-to-PostIMC registrations, which shows the distances of grid points before versus after registration. The linear registrations for

PostIMC-to-PreIMC and PreMSI-to-PostMSI both have low MLDs of around 2 $\mu$m. However, landmark detection for PostMSI and the matching of landmarks to PreMSI is more challenging due to the MALDI-MSI Matrix on top of the tissue, which alters its appearance and makes it more difficult to obtain enough landmarks for evaluation. The highest uncertainty in the

registration comes from the PreIMC-to-PreMSI step that compares non-linearly registered adjacent slices, which has been shown to be difficult (previously reported MLDs of around 20–30 $\mu m$;[28,29]). Because of these inherent difficulties, the precision of 63 (19%) out of 340 pairwise PreIMC-to-PreMSI registrations in our case study (68 samples, 5 registration steps) is either missing or very low, requiring manual (visual) inspection. Example images of the individual co-registration steps are shown in Supplementary Figs. S4–S9.

To further demonstrate the precision of the IMC-to-PostIMC co-registration, a separate experiment was conducted using 10 regions from 3 samples (see Methods) where after (lower-laser-power) IMC, an H&E stain was obtained to compare nuclei locations. The MLD between centroids of overlapping nuclei from IMC and H&E across regions is around 2 $\mu m$, similar to those in the case study (Fig. 2b; DICE coefficient is ~ 0.4: Supplementary Figs. S10, S11).

Manual registration of a (high-resolution) H&E stained image to a (low-resolution) PostMSI slide scan typically involves selecting two matching landmarks on each image[30,31]. To investigate this common practice, we performed this procedure using a slide from the case study using a high-resolution H&E (1$\mu m$ pixel size) and low-resolution PostMSI scan (approximately 10 $\mu m$ pixel size) (Supplementary Fig. S12, see Methods). Additionally, the same data was registered using the automatic approach described above. A large number of manually selected landmarks were used to calculated the performance of registration in this setting. The automatic registration achieved a smaller MLD between landmarks (17.1$\mu m$) compared to the manual registration (44.9 $\mu m$; see Fig. 2c). If on the other hand two high resolution images are used, as in the case study, the median distance decreased further to 2.4 $\mu m$. This suggests that high-resolution microscopy images and automatic co-registration (e.g., methods described in the ACROBAT challenge[29]) can improve the quality of co-registration.

To evaluate the precision of the automatic and manual registration of MSI-to-PostMSI, distances between MSI pixel locations and centroids of MSI ablation marks were calculated after registration. Similar median distances of around 2.0–2.5 $\mu m$ were obtained (Fig. 2d, see Methods). Furthermore, as expected, repeated registration of the same images resulted in small inconsistencies using manual registration, contrary to the automatic (deterministic) registration.

After all preprocessing, co-registration and filtering steps (Box A, B1-B2 and C1-3 in Fig. 1), the quantitative MSI and IMC modalities can be transformed into a common space via the chain of inferred transformations, leading to a set of high dimensional images containing both IMC-derived cell type information and MSI-derived m/z intensities (D in Fig. 1).

**Summarizing analyte-cell type associations using pixel-level and slice-level linear models**

The next step in the workflow is to link co-localization of cell types (via IMC) with the analyte occurrences measured (MSI) (Box E in Fig. 1), noting that the challenges arise due to different resolutions. Formally, we may wish to test for analyte-cell type associations (i.e., for every m/z value of interest, we want to find associations with presence of cell types). Given the magnitude of data for a larger experiment, it is necessary to first fit a pixel-level model for each tissue section to derive slide-level associations and in a second step, fit a sample-level model across a set of samples measured.

First, a simultaneous autoregressive model (SAR) can be fit (separately for each m/z value) for all pixels of each sample (TMA core) with the following specification:

$$\mathbf{y_j} = \rho_j \mathbf{W_j} \mathbf{y_j} + \mathbf{X_j} \beta_j + \epsilon, \qquad (1)$$

where for sample $j$ with $n$ pixels $\mathbf{y_j} = \{y_{j1}, \ldots, y_{jn}\}$ represent the (normalized) MSI pixel intensities for a specific mass-to-charge (peak), $\rho_j$ is a parameter representing the strength of spatial dependency between neighbouring pixels, $\mathbf{W_j} \in \mathbb{R}^{n \times n}$ the (row-standardized) spatial weight matrix (e.g., 1 if neighbour, 0 otherwise), $\mathbf{X_j} \in \mathbb{R}^{n \times (c+1)}$ represents the per-cell-type areas for a given MSI pixel (normalised by the total area of a MSI pixel) derived

from the IMC segmentations or directly the mean intensity of a IMC channel (in an MSI pixel), $\beta_j = \{\beta_{j0}, \ldots, \beta_{jc}\}$ the regression parameters of interest, $\epsilon_i$ are Gaussian errors, $\mathcal{N}(0, \sigma^2)$. Parameters in each $\beta_j$ vector represent analyte-cell type associations, summarized over the measured region of tissue, between a specific MSI peak to cell type occurrence. Because multiple MSI peaks are of interest, the analyte-cell type summaries can be represented as a matrix of quantitative information that will be used in the second fitting step.

The second step involves fitting a (mixed, if multiple samples per patient) linear regression model for each m/z value using the estimated coefficients $\beta \in \mathbb{R}^{m \times (c+1)}$ as response with $m$ total samples and sample-specific metadata ($\mathbf{Z} \in \mathbb{R}^{m \times z}$) as covariates with $z$ the number of covariates.

$$\beta = \mathbf{Z}\gamma + \epsilon \qquad (2)$$

where $\gamma \in \mathbb{R}^z$ are the regression coefficients to be estimated and $\epsilon$ the errors. The weighted least squares is evaluated with the inverse of the (estimated) variances of the $\beta$ as weights. The outcome for each m/z value are sample-specific associations $\beta_j$ and potentially, across-condition associations $\gamma$.

To investigate the behavior of this model in an idealized case, a small simulation study was conducted (see Methods). In short, results show that the correct values are estimated with the caveat that estimating the true associations becomes worse with larger registration errors (larger translation error) and with more extreme cell area errors, in terms of increased (estimated) variance of coefficients and attenuation bias towards zero (Supplementary Fig. S14). For the case of finding different strength of associations between conditions, a similar pattern is observed (Supplementary Figs. S15, S16). The simulation study indicates that higher registration precision gives better analyte-cell type association estimates, highlighting that rigorous registration evaluation is crucial to the integration of MSI and IMC modalities.

**Enhancing consistency and statistical power: a reanalysis of public IMC-to-MSI data**

Next, we demonstrate our pipeline on an arms-length dataset. To this aim, we reanalyzed a dataset of IMC (1 $\mu m$ pixel size) and lipid-MSI (5 $\mu m$ pixel size) generated by Nunes et al.[12] on the same tissue section, for which the authors calculated per-cell m/z intensities based on cell-to-MSI pixel overlaps. We highlight that the translation-only manual co-registration employed may be suboptimal because rotations or shears do occur, and that the use of Euler or affine transforms is highly recommended. Our reanalysis using automatic intensity-based registrations (Euler transformation) resulted in modestly better agreement of MSI and IMC intensities than the manual registration, based on visual inspection and increased structural similarity index measures (Fig. 3a, Supplementary Fig. S17). Furthermore, for most samples, the mean displacement between pixels from the manual to automatic approach was larger than the 5 $\mu m$ MSI pixel size (Supplementary Fig. S17). Although these are subtle shifts, they are propagated to downstream analyses, visible as cell-area-dependent changes of m/z intensities between different registrations (Fig. 3b) that also depend on the MSI intensity (Fig. 3c). After modeling the analyte-cell type associations from MIMIC (as described above), estimates are largely similar across registrations, but the corresponding p-values are in general slightly lower for the Euler registration (Supplementary Figs. S18, S19). Zooming in at plasma cells annotated as either CD138+ or CD138-, the modeling results for the automated registration are more similar between the subpopulations than for the manual approach (Fig. 3d, Supplementary Fig. S20), suggesting indirectly that better (automated) registrations can improve the robustness of results.

**Experimental Validation: Artificial Tissue as a Proof of Principle**

To further assess the performance of finding analyte-cell type associations in a real dataset with a ground truth, a synthetic tissue was created. In short, three distinct cell lines (Huh7, AC16 and Mel624) were separately labeled

## Case study coregistration validation

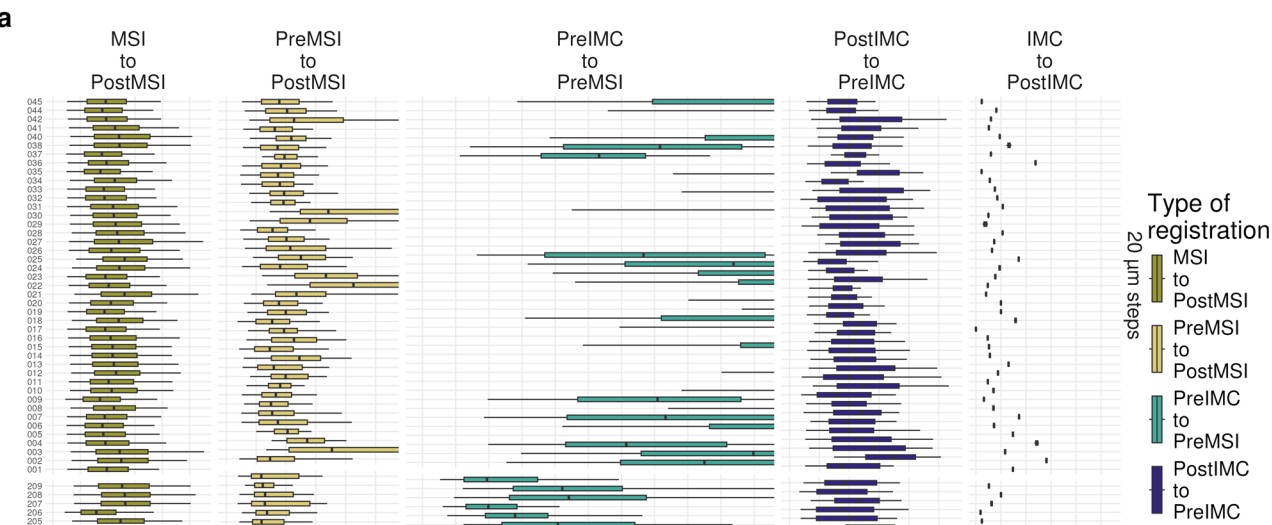

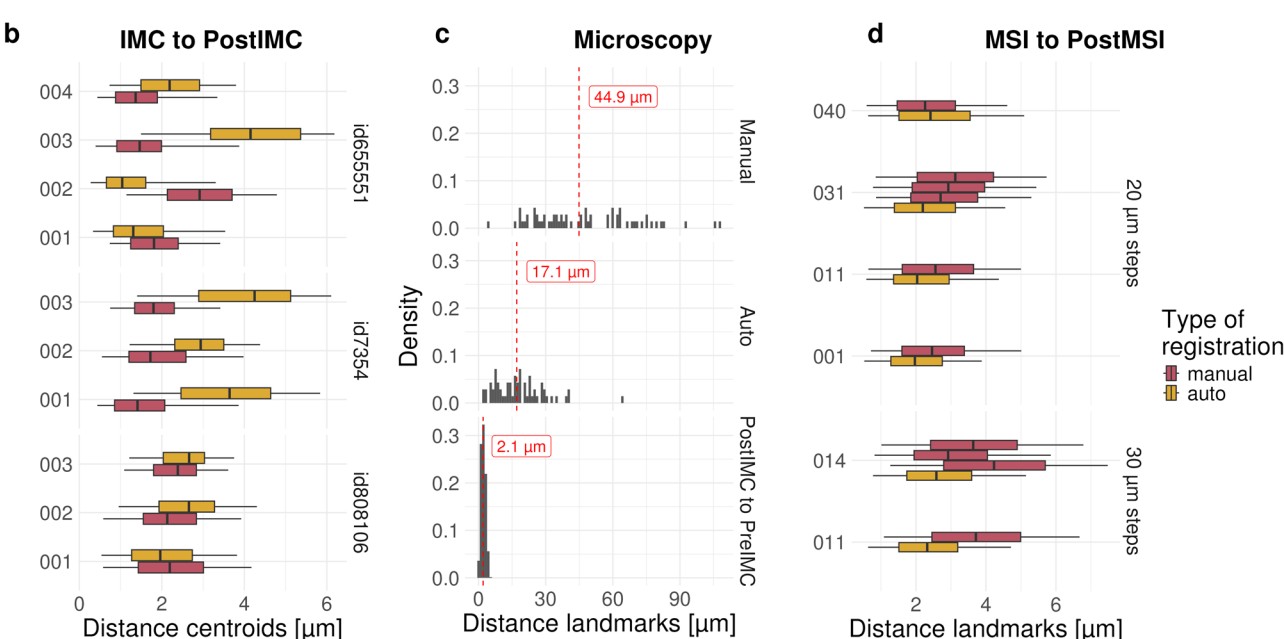

**Fig. 2 | Evaluation precision of registrations. a** Distances between automatically detected landmarks for the five co-registration steps in the case study: MSI-to-PostMSI, PreMSI-to-PostMSI, PreIMC-to-PreMSI, PostIMC-to-PreIMC, IMC-to-PostIMC. Results with less than 100 landmarks were excluded. **b** IMC-to-PostIMC, distances for comparison of cell nuclei position and overlap from IMC and H&E for manual and automatic registration. **c** Distances between manually selected landmarks for manual (top) and automatic (middle) registrations using low resolution images. Distances between landmarks of high resolution images and automatic registration (bottom). **d** MSI-to-PostMSI, distances between MSI pixel locations and centroids of MSI ablation mark for manual and automatic registration for a subset of samples from case study.

with distinct isotopes ($^{115}$In, $^{194}$Pt and $^{196}$Pt), followed by spotting on a slide either in-isolation (one replicate per cell line) or as a mixture (two replicates) (Fig. 4a, see Methods). Lipid-MSI (10 $\mu m$ resolution) and IMC (1 $\mu m$ resolution) were run serially on the same slide. MIMIC's complete workflow was run on all samples to align IMC-MSI modalities and derive the lipid-cell type associations. Ground truth lipid-cell type associations were derived

from global mean differences between the in-isolation cell line samples (see Methods; Supplementary Fig. S21). For example, m/z and isotope (cell line) intensities qualitatively show associations: m/z 728.52 and Mel624 for both the in-isolation and the mixture samples (Fig. 4b). Quantitatively, the observed MSI-pixel-level associations between m/z 728.52 and Mel624 are shown in Fig. 4c. Given the derived ground truth, the lipid-cell type

association estimates from the pixel-level models were split into two groups: those higher expressed in Mel624 (Fig. 4d), or those higher expressed in AC16 (Fig. 4e). Cell line Huh7 was omitted from further analysis because of low [115]In signal. Especially the Mel624 cell line shows, as expected, positive associations for in-isolation and mixture samples in Fig. 4d that are generally larger than those for AC16. For AC16, the coefficients are on average also larger than Mel624 but are often negative and thus smaller than expected. Overall, the results are consistent with the ground truth, although the measurements are noisy and estimated standard errors of the lipid-cell type associations can be rather high. A sensitivity analysis (i.e., increasing registration errors), shows a convergence of the estimates towards zero (Supplementary Figs. S22, S23. These observations further highlight the importance of careful registration and replicates for more robust results.

### MIMIC identifies cell type-glycan associations in the liver

We now return to the data of our case study, which involves two TMAs containing a total of 68 liver explant cores from patients with varying stages of metabolic dysfunction associated steatotic liver disease (MASLD) (see Methods). For each TMA, adjacent tissue slices were prepared (and processed independently) for IMC and glycan-MSI. As described previously, microscopy slide scans were taken before and after IMC or MSI. The first TMA contained cirrhotic liver tissue, analyzed with an MSI resolution of $30\,\mu m$. The second TMA included samples with either hepatocellular carcinoma or adjacent liver tissue, analyzed with an MSI resolution of $20\,\mu m$. IMC images were all acquired with a spatial resolution of $1\,\mu m$, using a panel that included markers for the identification of immune and liver cell types as well as those for cell segmentation.

To obtain an integrated dataset, we ran MIMIC co-registration on the IMC and MSI modalities. Per-MSI pixel summaries, such as the number of pixels that a single cell overlaps with, show that cells overlap with more MSI pixels if the MSI pixel size is smaller (Fig. 5a). Still, for MSI pixel sizes of 20 and $30\,\mu m$, most cells only overlap with a single pixel. From the opposite viewpoint, the average number of cells that overlap with an MSI pixel increases for larger and decreases for smaller pixel sizes, from 4.2 for $30\,\mu m$ pixels to 2.5 for $20\,\mu m$ pixels (Fig. 5b). Moreover, the distributions of the "filled-with-cell" area of MSI pixels normalized by the total pixel area are, as expected, similar for $20\,\mu m$ and $30\,\mu m$ (Fig. 5c). The overlap summaries also depend on the type of tissue, as seen for the densely-packed artificial tissue, where observed cell-MSI overlaps show a large proportion of MSI pixels filled with cells (Supplementary Fig. S24).

After co-registration, both IMC and MSI were processed separately to obtain cell type abundance and glycan intensities, respectively. The IMC image processing (see Methods) recovered all expected cell populations (Supplementary Figs. S25, S26). For the glycan-MSI data, after peak binning and normalization (see Methods), per-sample "bulk" mean m/z profiles were constructed and compared with cell type proportions per sample (Supplementary Fig. S27b). As an example of our results, we found that m/z 1809.639 was negatively associated with Hepatocytes and positively associated with immune cells for both TMAs, while m/z 1743.581 showed the opposite directionality (Supplementary Fig. S27b), matching previously-described associations[32]. Still, a per-sample aggregation resulted in only a few detected analyte-cell type associations (Supplementary Fig. S28).

A better avenue is to additionally take into account the spatial position of cells, as obtained from the co-registered IMC images, to better delineate associations with glycans. In liver tissue with advanced fibrosis two distinct spatial domains can be identified: one with nodules of regenerating hepatocytes, which mainly contains hepatocytes and one with bridging fibrous septa, which is composed by fibroblasts, collagen, and leukocytes[33]. Based on this tissue architecture, per sample and domain "pseudobulk" mean glycan profiles were constructed and tested for differences between the domains. Results show several glycans that are associated with either of the two domains (Supplementary Fig. S29), as previously reported in the literature[32].

Both aggregation approaches, at the sample- or domain-level, potentially average out useful information that can be retained only if directly looking at the pixel-level values. For instance, m/z 2067.684 clearly correlated with hepatocytes at the pixel-level for a single sample (Fig. 6a-d) and also across all samples (Fig. 6e), but did not at the bulk (aggregated per sample) or pseudobulk (aggregated per sample-domain) level (Supplementary Figs. S28, S29).

To obtain a global overview of glycan-cell type associations, all m/z values were systematically tested for associations with cell types. A subset of these associations (pixel-level model) and corresponding FDR corrected p-values (slide-level model) are shown in Fig. 6e (full results can be found in Supplementary Fig. S30).

While the estimates of association do not seem to depend on the abundance of a cell type (top annotation versus estimates in heatmap in Fig. 6f), the -log10 FDR corrected *p*-values are, as expected, on average negatively correlated with the proportion of missing m/z values or in other words, positively correlated with number of observations (Supplementary Fig. S31). This observation is in accordance with the '4S-criteria for performance' and emphasizes the dependence between sensitivity and spatial resolution, both of which influence the number of non-missing observations (pixels).

Overall, we observed two distinct groups: hepatocytes plus CD16+ Kupffer cells versus all other cell types, resembling the pseudobulk-level results, i.e., the difference between hepatocyte-rich and inflamed regions (Supplementary Fig. S29). Previous research has shown that high-mannose glycans are predominantly expressed in the hepatocyte-rich areas of the fibrotic liver[32,34], including those represented by m/z 1419.475, 1581.528, 1743.581 and 1905.634, which all have significant, positive estimates of association for hepatocytes in our dataset (black boxes in Fig. 6e, Supplementary Fig. S30). Defining a reference dataset for the N-glycan profiles of immune cells is more challenging. However, in our dataset, we find a positive association for glycans (black boxes in Fig. 6e, Supplementary Fig. S30) that were previously shown to be preferentially expressed on an immortalized T-cell-leukemia cell line (Jurkat cells)[35]. A sensitivity analysis reveals that estimates tend to converge towards zero for larger registration errors for cell types with large local spatial heterogeneity (e.g., CD8 cells), while for cell types with smaller local spatial heterogeneity (e.g., Hepatocytes), the estimates are more stable (Supplementary Fig. S32).

## Discussion

High-dimensional spatial technologies that allow the identification of single cell types, transcriptomes, lipids, glycans and metabolites are pushing the boundaries of tissue analysis allowing for a comprehensive interrogation of tissue heterogeneity. However, the integration of data from multiple spatial modalities is difficult when the spatial resolution is different. Especially when combining MALDI-MSI, which generally does not reach single-cell resolution, with single cell-based technologies, considerable challenges remain. To overcome these, we developed a (semi-)automated registration workflow integrating MSI and IMC starting from raw images, with possible manual interventions. MIMIC allows the precise registration of MSI and IMC data with the use of intermediate microscopy images, quantitative and qualitative measures of the registration precision and the per-pixel integration for downstream analysis. The ability to combine IMC and MSI from either a single slice or consecutive slices allows flexible experimental setups. Furthermore, the registration of MALDI-MSI pixels to ablation marks allows the use of lower resolution MALDI-MSI for applications where high sensitivity is required or acquisition speed is a limiting factor. The up to 40 channels in IMC is ideal to finely distinguish cell types and the relatively low resolution compared to fluorescent imaging is sufficient for the integration with the even lower resolution MALDI-MSI.

To validate the robustness and utility of MIMIC, we applied the workflow across three settings: a synthetic tissue with three distinct cell lines, a re-analysis of publicly available IMC-MSI data, and a human liver tissue microarray from a large cohort of patients diagnosed with MASLD. In the re-analysis of the dataset generated by Nunes et al., we showed the importance of precise co-registration on robustly detecting analyte-cell type association. However, missing microscopy images meant an alternative co-registration based on MSI and IMC intensities needed to be employed and

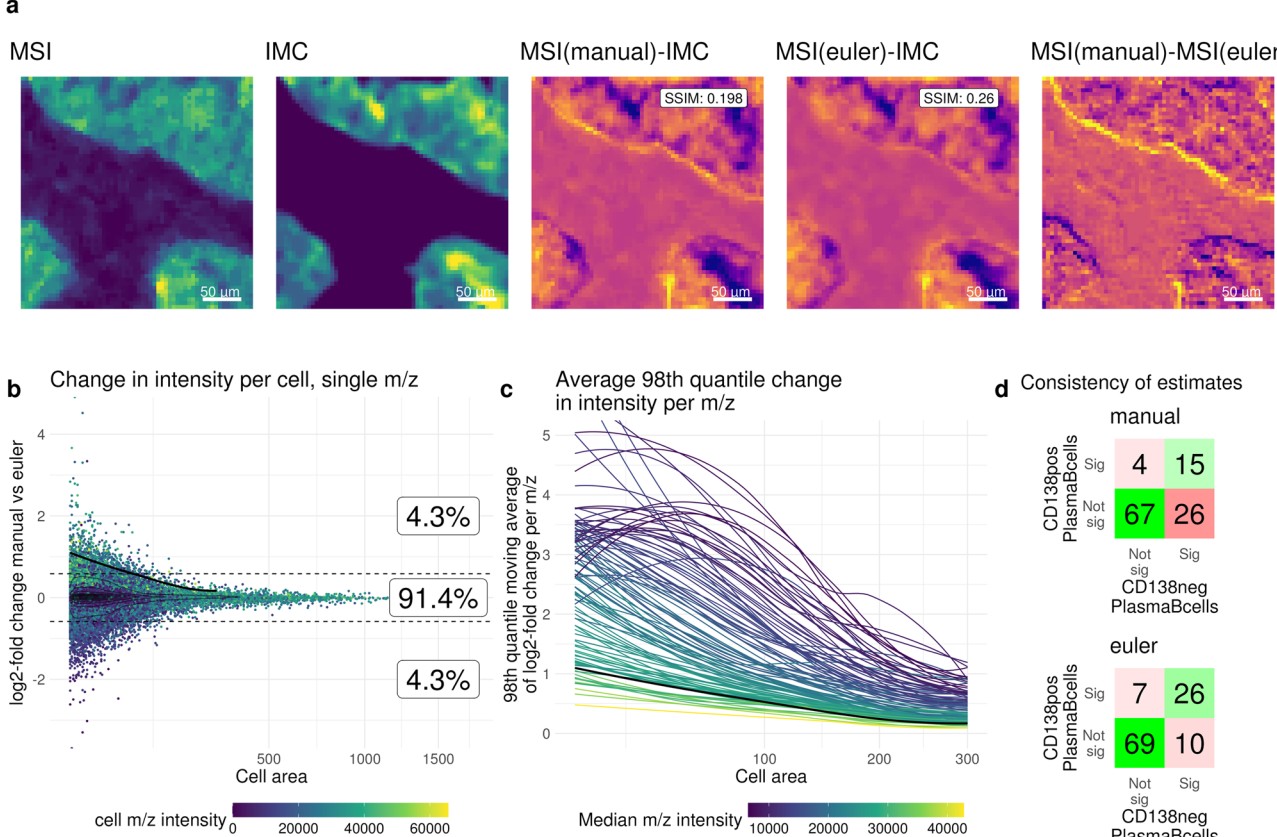

**Fig. 3 | Reanalysis of data from Nunes et al.[12]. a** Example image section displaying differences in registration. From left to right: MSI intensity image (3 channels combined), IMC intensity image (Keratin), difference of MSI (manual registration) and IMC, difference of MSI (Euler registration) and IMC, difference of MSI (manual registration) and MSI (Euler registration). SSIM: structural similarity measure. **b** cell level MSI intensity differences between manual and Euler registration, log2-fold change of example MSI channel of manual vs. Euler registration with the cell area on the x-axis. Boxes show the percentage of cells with a log2-fold change higher than $log_2(0.5)$, lower than $- log_2(0.5)$ or in between. The black line is the moving average of the 98th quantile. The dashed lines represent the $log_2(0.5)$ and $- log_2(0.5)$ thresholds. **c** Each line represents an MSI channel showing the moving average of the 98th quantile of the log2-fold change across a range of cell areas. The black line is the same as in (**b**). **d** Result of associations between CD138 positive (pos) or CD138 negative (neg) Plasma B-cells and m/z intensities. Shown are the number of significant associations according to FDR corrected p-values (threshold: 0.01) for manual (top) or Euler (bottom) registration.

direct registration errors could not be calculated. In the artificial tissue experiment, known lipid-cell line relationships were successfully recovered under controlled conditions. In the liver tissues, MIMIC detected both expected and novel glycan-cell type associations. Our analysis revealed associations that matched previously described relationships of high-mannose glycans with hepatocyte-rich domains, while also uncovering new associations with immune cell subsets. Importantly, compared to bulk or pseudobulk aggregation approaches, only pixel-level modeling allowed cell-type-specific delineation with analytes, highlighting the importance of precise data integration.

Despite the high level of automation and robustness, several limitations remain. Specifically, while measuring consecutive slices can be an advantage experimentally, integrating the images relies on the assumption that tissue is very similar across thin, adjacent slices. Still, considering the tissue thickness of $5\,\mu m$ and MSI resolution of $10–30\,\mu m$, measured MSI signals on consecutive slices should be more similar to each other than to neighboring pixels on the same slice. Furthermore, precisely aligning adjacent slices remains a challenge due to biological differences across sections, requiring manual inspection and exclusion of samples with large differences across sections. Furthermore, microscopy image acquisition artifacts, such as slightly-different planes or possibly-altered samples, might require image transformations with larger degrees of freedom for successful registration. Additionally, although the spatial dislocation observed in MALDI-MSI is at least partially dealt with in the modeling step, further quantification and

correction might help to increase signal. Furthermore, the lack of a generally-accepted (pre-)processing workflow for MSI data may result in irreproducible results depending on the chosen steps. Looking ahead, several extensions of MIMIC are possible. Incorporating H&E staining prior to IMC acquisition could improve the validation of the precision of registration[36]. Furthermore, pixel-level classification methods, such as PIXIE[37], could be an alternative to classical cell type identification based on segmentation, removing an IMC processing step with relatively high variability. Furthermore, since the registration workflow is structured around the intermediate microscopy images, extending MIMIC to include other modalities, such as spatial transcriptomics, is possible if the respective modality can itself be registered to a microscopy image. As such, for successful co-registration, there are some requirements on the microscopy image itself, such as similar resolution and similar visible spatial structures. Ideally, additional modalities would be co-registered automatically (for reproducibility) and evaluate the precision of co-registration to ensure robustness. Furthermore, if modalities are acquired over more than two slides, the target microscopy image needs to be carefully chosen to reduce co-registration errors. In conclusion, MIMIC enables high-precision, robust and reproducible integration of MSI and IMC, overcoming previous limitations in multimodal spatial analysis. Its application to both synthetic and clinical tissues highlights its versatility and potential to advance spatial systems biology, disease characterization, and tissue-based diagnostics in research and clinical settings alike.

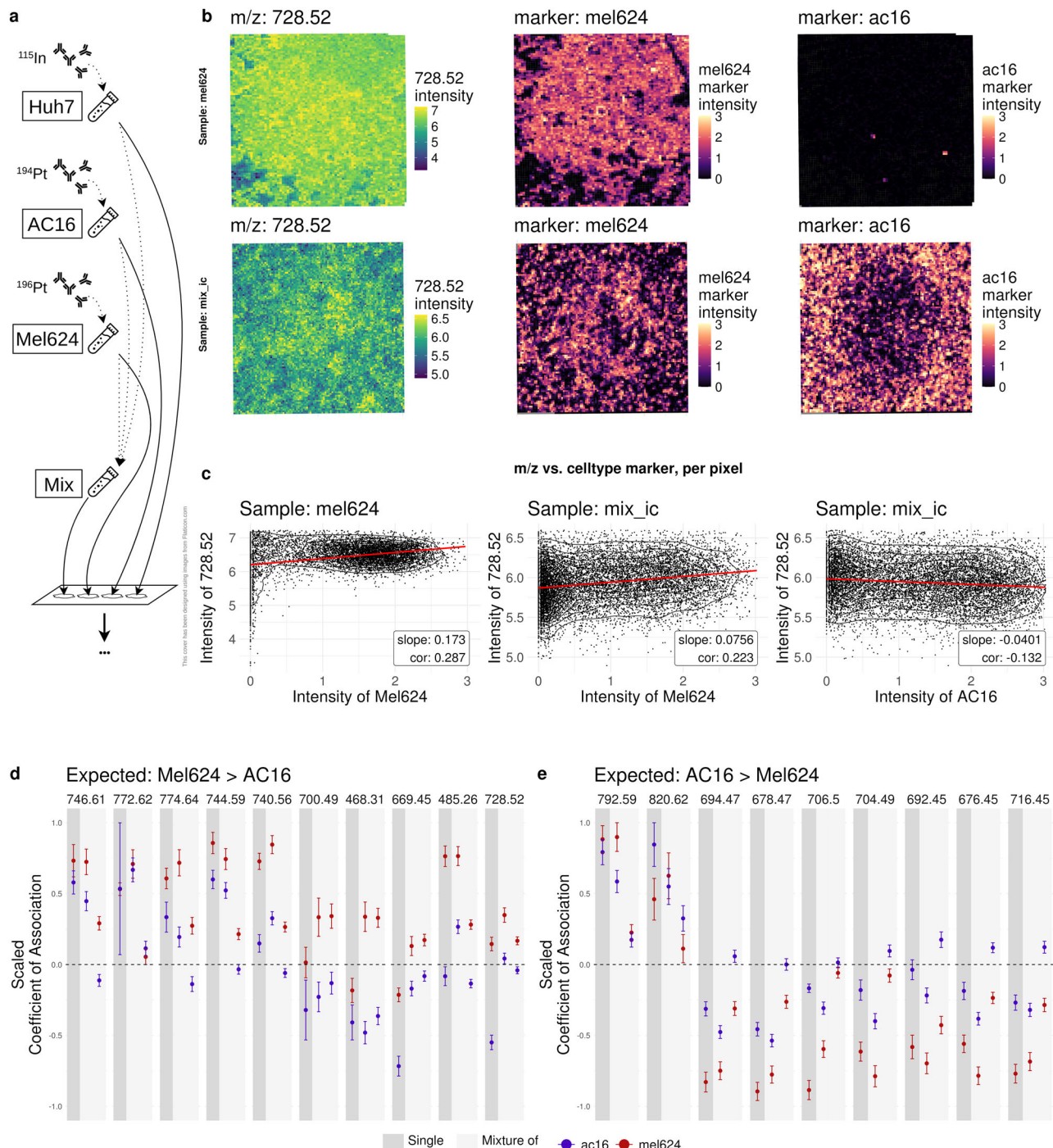

**Fig. 4 | Results of the workflow with the artificial tissue. a** Experimental setup. Three different cell lines where labeled with different isotopes and spotted on a slide, either as a homogeneous or a mixture. **b** Visualization of example lipid (images in left column) and cell type markers (images in middle and right columns) for a sample containing only Mel624 cells (top row) and a mixture of cells (bottom row). **c** Intensity of lipid vs. cell type marker intensities, same order as in (**b**). The slope of fitting a linear model as well as the pearson correlation coefficient are shown

additionally. **d** Estimates of association for cell line Mel624 and AC16 with various lipids expected to be higher expressed in Mel624. Results for both the homogeneous and the mixture samples are shown. **e** same as (**d**) but expected higher expression in AC16. Estimates and error bars (standard deviation) for (**d, e**) are linearly scaled between −1 and 1 for simpler visualization.

## Methods
### Artificial Tissue Creation
Human cell lines Huh7 (hepatocellular carcinoma), AC16 (human cardiomyocyte), and Mel624 (melanoma) were cultured under ATCC standard conditions and harvested by trypsinization. Post-harvest, cells were washed with PBS with 1% BSA and 0.1% EDTA. Cells were stained with previously validated anti-$\beta$2-microglobulin (B2M) metal-conjugated antibodies

(Clone A17082A BioLegend): Huh7 cells with B2M Indium-115, AC16 cells with B2M Platinum-194, and Mel624 cells with B2M Platinum-196. Staining was performed by adding 2 $\mu L$ of the respective antibody to 2 mL of FACS buffer containing the cell suspension, followed by incubation at 4 °C for 2.5 h. After staining, cells were washed twice with phosphate-buffered saline (PBS) and then simultaneously fixed while staining DNA using Cell-ID Intercalator Ir (Standard BioTools PN: 201192B) diluted 1:1000 in

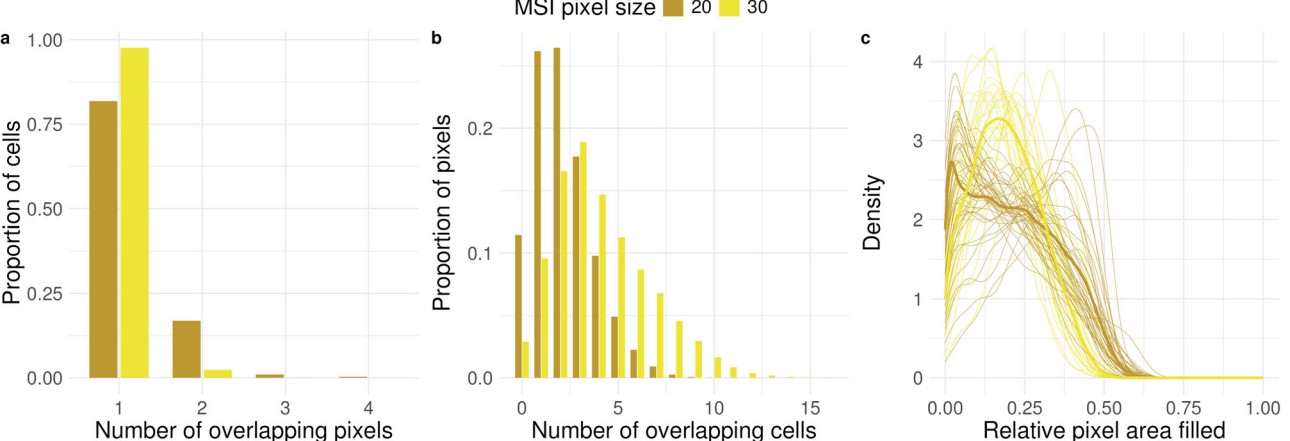

**Fig. 5 | Distributions of overlaps between cells and MSI pixels for the case study. a** Distribution of number of pixels a cell overlaps with. **b** Distribution of number of cells an MSI pixel overlaps with. **c** Distribution of relative area filled by cells per MSI pixel.

Maxpar Fix and Perm Buffer (Standard BioTools PN: 201067) for thirty minutes. After an additional PBS wash, cells were countedyielding between 2.5 and $3.5 \times 10^6$ cells per mL. The labeled cell were then mixed at a 1:1:1 ratio and $10 \, \mu L$ aliquots were spotted onto a Poly-L-Lysine coated ITO slide (Delta Technologies GC-811N-S115). The spotted samples were incubated on an orbital shaker at 150 rpm at room temperature for 1 h. Following incubation, samples were vacuum-dried and fixed with 4% paraformaldehyde (PFA) for 30 min. Finally, cells were washed with high-performance liquid chromatography (HPLC)-grade water to remove residual fixative.

## Tissue processing

This study utilized liver biopsy specimens obtained from patients treated at the Medical University of South Carolina (MUSC) between January 1, 2012, and June 1, 2021, under IRB protocol PRO111306. Specimens were collected as part standard care procedures such as diagnostic biopsy or transplantation. Retrospective review of patient medical records gathered demographic, clinical, and laboratory data, with inclusion criteria based on pathology reports citing the Nonalcoholic Steatohepatitis (NASH) Activity Score (NAS). Chart reviews confirmed that patients were positive for metabolic syndrome and negative for viral hepatitis or heavy alcohol use. Outliers in age or BMI were excluded from the cohort. All specimens were de-identified, assigned a unique numeric code, and stored securely to protect patient confidentiality. The tissue microarrays (TMAs) were constructed by the Biorepository and Tissue Analysis Shared Resource at the Hollings Cancer Center.

## Imaging Mass Cytometry Sample Preparation and Data Acquisition

Prior to image acquisition, Imaging Mass Cytometry (IMC) parameters—including antibody dilutions, iridium-intercalator solution concentrations, and image acquisition strategies—were optimized. Formalin-fixed, paraffin-embedded (FFPE) liver tissue sections of $5 \, \mu m$ thickness were heated at 60 °C for 1 h to facilitate wax removal. Deparaffinization was performed using the following sequence of solutions: two washes in xylenes for 3 min each, two washes in 100% ethanol for 1 min each, one wash in 95% ethanol for 1 min, one wash in 70% ethanol for 1 min, and two washes in HPLC-grade water for 3 minutes each.

Heat-induced epitope retrieval (HIER) was conducted using Tris-EDTA pH 9 HIER Buffer (Abcam, ab93684) by incubating the sections at 95 °C for 20 min in the Decloaking Chamber™ NxGen (BioCare Medical). The tissue sections were encircled with the ImmEdge® Hydrophobic Barrier PAP Pen (Vector Laboratories), and nonspecific binding was blocked by incubating with TBS containing 0.1% Triton X-100 and 3% BSA at room temperature in a humidity chamber for 1 hour. An antibody master mix was prepared by diluting antibodies to empirically determined concentrations in TBS containing 0.1% Triton X-100 and 1% BSA. After the blocking incubation, the blocking buffer was gently removed, and the antibody master mix was applied to the tissues. The sections were incubated overnight at 4 °C in a humidity chamber.

Following incubation, the slides were washed in Coplin jars for 5 min: twice with PBS containing 0.2% Triton X-100, followed by two washes with PBS alone. Subsequently, Intercalator-Ir (Standard BioTools) was diluted 1:1000 in PBS and incubated on the tissues at room temperature in a humidity chamber for 30 min. The slides were then washed twice in HPLC-grade water for 5 min each and dried in a vacuum desiccator for at least 1 h.

Approximately $1000 \, \mu m \times 1000 \, \mu m$ square regions of the tissue sections were analyzed using the Helios and Hyperion Tissue Imager (Standard BioTools). The instrument was tuned and calibrated using the 3-Element Full Coverage Tuning Slide (Standard BioTools) to ensure that the 175Lu counts exceeded 750. Data acquisition was performed with a step size of $1 \, \mu m$, a laser frequency of 200 Hz, and a laser energy setting of −1 dBa. Regions of interest (ROIs) were selected based on tissue histology, and image visualization was conducted using MCD Viewer version 1.0.560.6 (Standard BioTools).

## MSI measurement: On-tissue sialic acid stabilization

FFPE tissue slides were heated at 60 °C for 1 h to melt the paraffin, followed by dewaxing in xylenes and rehydration through a series of graded ethanol solutions (100%, 95%, 70%) and HPLC-grade water. The tissue slides were then dried in a vacuum desiccator for 30 minutes. To differentiate structural isomers containing sialic acid residues, we performed a differential labeling of $\alpha 2,3$- and $\alpha 2,6$-linked sialylated glycans using the workflow described in ref. 38. Briefly, for the first reaction, 200 $\mu L$ of the AAXL amidation solution, containing 1-(3-dimethylamino-propyl)-3-ethylcarbodiimide (EDC), 1-hydroxybenzotriazole (HOBt), dimethylamine, and dimethyl sulfoxide (DMSO), was applied directly to the tissue surface, and a glass coverslip was placed to ensure even reagent distribution. Slides were incubated at 60 °C for 1 h in a sealed container. After incubation, the coverslip was removed, and the reaction solution was aspirated. The tissue was washed three times with $400 \, \mu L$ of DMSO, followed by vacuum aspiration. The second reaction solution, consisting of $200 \, \mu L$ of a $300 \, \mu L$ propargylamine/$700 \, \mu L$ DMSO mixture, was applied to the tissue, and a fresh coverslip was placed. This second reaction was incubated for 2 h at 60 °C in a sealed container. After incubation, the coverslip was removed, and the tissue was washed twice with 100% ethanol for 2 min, twice with Carnoy's solution (60% EtOH, 30% chloroform, 10% acetic acid) for 10 minutes, followed by two more ethanol washes and a brief wash in 0.1% TFA in ethanol. The slides were finally rinsed in HPLC-grade water and desiccated for 5 min to overnight.

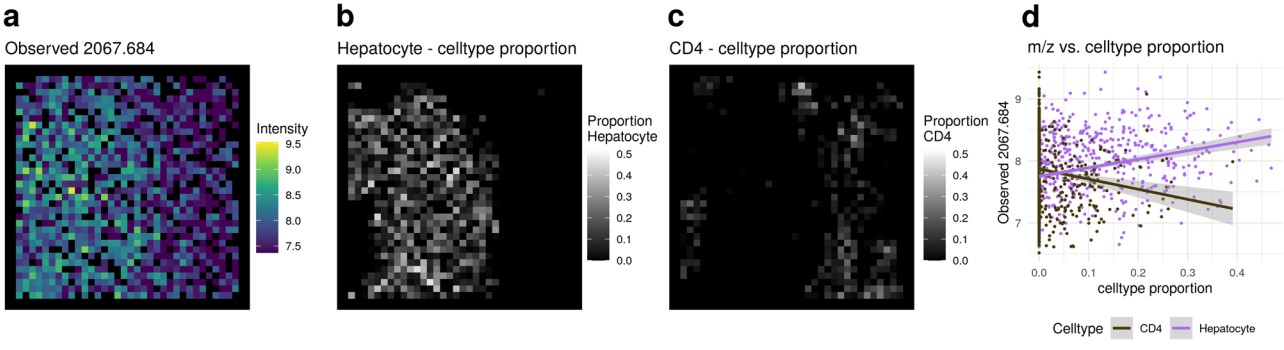

**Fig. 6 | Results of statistical analysis. a–d** Example of fit for single sample and single m/z value. **a** observed m/z value. **b** cell type proportion of Hepatocyte. **c** cell type proportion of Myofibroblast. **d** cell type proportion vs observed m/z value. **e** Estimates of association of single m/z value with cell types for individual samples.

**f** Estimates of association of cell types with m/z across samples. The left annotation shows the mean m/z intensity per sample. The top annotation shows the cell type proportion per sample. Only the top 5 m/z values per cell type according to FDR with FDR0.05 are shown. Black boxes represent known associations.

Following sialic acid derivitization, the antigen retrieval process was performed using 10 mM citraconic anhydride buffer (pH 3) at 95 °C for 20 min in the Decloaking Chamber™ NxGen (BioCare Medical). PNGase F (PNGase F Prime, N-zyme Scientifics) was sprayed for 15 passes using an automated sprayer (M5, HTX Technologies) with the following parameters: temperature of 45 °C, pressure of 10 psi, flow rate of 25 μL/min, velocity of 1200, nozzle positioned 40 mm above the tissue,

and a 2.5 mm offset. The tissue was then enzymatically digested in a preheated humidity chamber for 2 hours at 37.5 °C. After digestion, CHCA matrix was prepared at 7 mg/mL in 50% acetonitrile and 0.1% TFA, and sprayed using the automated sprayer for 20 passes with the following parameters: temperature of 79 °C, pressure of 10 psi, flow rate of 50 μL/min, velocity of 1300, nozzle positioned 40 mm above the tissue, and a 3.0 mm offset.

## MSI measurement: MALDI-QTOF MSI Acquisition Parameters

Mass spectrometry imaging was performed on a timsTOF fleX MALDI-QTOF instrument (Bruker Daltonics, Bremen, Germany) operated in qTOF mode with the trapped ion mobility spectrometry (TIMS) device deactivated. All data were acquired in positive ion mode over an m/z range of 700–4000. The instrument was tuned with the following global settings: MALDI Plate Offset of 30 V, Deflection 1 Delta at 70 V, Funnel 1 and Funnel 2 RF at 450 and 500 Vpp, respectively, and Multipole RF at 500 Vpp. The collision cell was configured with 10 eV collision energy and 2900 Vpp collision RF, and quadrupole transmission was achieved with an ion energy of 5 eV and a low mass cutoff of m/z 700. The focus optics were set with a transfer time of 110 $\mu s$ and pre-pulse storage of 10 $\mu s$. The SmartBeam 3D laser was operated at 10 kHz. Mass calibration was performed prior to acquisition using ESI-L Tune Mix (Agilent Technologies, Santa Clara, CA, USA). For spatial resolution optimization, the Cirrhosis TMA was imaged using a 30 $\mu m$ raster, with a ScanSizeX/Y of 20 $\mu m$ and laser power of 40%, applying 200 laser shots per pixel. The HCC TMA was acquired using a 20 $\mu m$ raster, with a ScanSizeX/Y of 12 $\mu m$ and a higher laser power of 50%, with 400 laser shots per pixel. All scans were performed with the beam scan activated and laser parameters adjusted to optimize ion signal and spatial fidelity. Additional instrument parameters are listed in Supplementary Table S3.

## Brightfield Slide Scanning

Bightfield scanning was performed using a Nano-zoomer 2.0-RS high resolution slide scanner (Hamamatsu) at 40x magnification. Automated calibration was performed before each scan.

## Overview of analysis workflow

The presented workflow has been designed to work with Tissue Micro Array (TMA) slides which has the benefit that many samples, including both biological and technical replicates, can be placed on the same slide allowing consecutive data collection which potentially reduces technical artifacts[39]. The required data are high resolution brightfield optical slide scans before and after taking MSI or IMC measurements plus the MSI and IMC measurements themselves.

Figure 1 b shows the overview of the whole analysis (See also Supplementary Fig. S1). Starting from the raw input data including the two data modalities IMC and MSI plus the microscopy slide scans the data is processed in multiple steps. The MSI and IMC data can be preprocessed and analyzed independently of each other (A and B1-2 in Fig. 1b). The co-registration (C1-3 in Fig. 1b) of the modalities combines IMC and MSI into a common frame of reference (D in Fig. 1b) making joint analysis and association testing possible (E in Fig. 1b).

## IMC

The processing of the IMC data is bundled together in a Snakemake[27] workflow starting from the raw .mcd files and ending with integrated and annotated data (See box IMC in Fig. 1b and Supplementary Fig. S33). Starting from the raw IMC data (in .mcd format) the multi-channel tiff images are created using steinbock[26]. The analysis can be broadly split into two parts, at the pixel level and at the cell level. For the pixel level, the three main steps are "Denoising", "Segmentation" and "Aggregation" all of which are run using steinbock (Fig. 1b B1). In "Denoising" hot pixels are removed (pixels that have a value that is 50 higher than the maximum of neighboring pixels are set to this threshold). In the next step, a simple denoising is performed by setting all pixels (of all channels) that have a value higher than a threshold (1.5 in this case) to 0. In "Segmentation" the cell masks are generated using Mesmer[40]. The input data is a two channel image where one channel is for cell nuclei and the other is for cell membrane. The two available DNA channels and the three specific cell membrane channel are each aggregated and used as nuclei and cell membrane markers respectively. In "Aggregation" geometric features are extracted (such as area, position, etc.) based on the previously created mask and the mean intensities per marker per cell are calculated.

Starting from "Cell level" the cell level data is used (Fig. 1b B2). From here on R[41] is used for analysis. First the intensities are converted to an R object (using imcRtools[26]) and then transformed using an arcsinh transform (with a cofactor of 1). Next, filtering of cells is performed with the following criteria: based on cell area (remove cells with area smaller than 10 pixels), based on location (remove cells that touch the border of the image), based on marker intensity (remove cells that have zero intensity for all cell marker channels or that have a cell type marker intensity that is higher than the median plus 5 times the median absolute deviation). Afterwards the cells are integrated (using Harmony[42]) and clustered (shared nearest neighbor graph and walktrap or leiden clustering using scran[43]). The obtained clusters are then manually annotated based on cell type marker intensities.

## MSI

Raw MALDI-MSI ".d" files are converted to "imzml" files in SCiLS Lab. The rest of the MSI specific analysis is conducted in R with the package Cardinal[44]. The imzml files are read in and first normalized by the total ion current (TIC). Next the spectra are aligned using a set of known reference values. The reference list for the glycans is the same as described previously[38]. Lastly, the spectra are binned at the reference values taking the peak height as intensity value. Furthermore the intensities are transformed using the logarithm with base 2. For the validation with the artificial tissue the peak list was generated based on possible lipids from Metaspace using the data bases SwissLipids and HMDB[45] and then filtered by differential intensities (t-test) of TIC-normalized m/z values between single cell line regions.

## Co-registration

The co-registration of the IMC and MSI modalities is done in multiple steps requiring additionally the microscopy images of the sample slides. More specifically two images per slide are needed, one before measurement and one after measurement. Or in case of acquisition of IMC and MSI on the same slide a total of three microscopy images are needed, before IMC, after IMC, after MSI. The following steps are implemented as a Snakemake workflow.

The first step involves mapping of the ablation marks in the PostMSI image to the MSI pixels. Two possibilities of how this is done are employed.

The first approach consists of multiple steps trying to register the images automatically (Supplementary Fig. S34). In short, for each tissue sample: First the PostMSI microscopy image is segmented to obtain the mask of the tissue samples (TMA cores). Next the MSI regions (i.e., connected components) are mapped to the corresponding tissues on the PostMSI microscopy image by registering the binary masks using SimpleITK, function `AffineTransform` with the metric to optimize set as correlation[46] (top panel in Supplementary Fig. S34). This step is required to subsequently register individual cores, the co-registration precision is not critical at this stage. Therefore, the shape of the MSI regions does not need to match the shape of the TMA cores, but the MSI region must be completely filled. Next the location of the MSI pixel ablation marks are detected individually for each MSI regions. This is achieved by processing the grayscale PostMSI image using a set of convolutions (median with square kernel, rank threshold with disk kernel, mean with cross kernel) and then creating a pixelwise weighted sum of the processed image with the grayscale PostMSI image. A threshold is then applied followed by measurement of connected components representing MSI pixels. The detected components are then filtered based on size, shape, distance and angle to neighbors together with a minimum sub-graph size. Grid search is applied to find the weight and threshold that optimize a score, composed of the sum of the normalized (between 0 and 1) number of detected points, four times the normalized (between 0 and 1) number of points at the edges of the tissue and two times the normalized (between 0 and 1) mean distance of points to the edge (bottom left panel in Supplementary Fig. S34). Next the obtained locations of MSI ablation marks on the PostMSI image are used to create a mask that can be registered to the MSI mask[46]. Another grid search strategy is employed next to find transformation parameters (x and y translation, rotation) that optimize a score based on the proportion of MSI ablation

marks in the MSI mask, number of matches of MSI ablation marks to MSI pixels, mean weighted distance of matched points (weight approximately inversely proportional to the density of points). Concave hulls based on the regular grid structure of the ablation marks are constructed for the MSI ablation marks and the MSI image taking into account the regular distances and angles between points. For all detected points on the boundary of the concave hull identifiers of different lengths are generated based on the sequence of angles to neighboring points (e.g., using 2 neighbors clockwise and 3 neighbors anti-clockwise leading to an identifier of "90,180,180,180,270,180,90"). Potential matching points are first filtered by the physical distance based on the initial transform from the previous step by setting a maximum allowed distance. Second, the identifiers are compared between ablation masks and MSI pixels. An optimal identifier length is determined by ensuring only unique matches and maximizing the number of matches and the proportion of covered border sections. The matched PostMSI ablation marks are then registered to the matched MSI locations using coherent point drift[47,48] (bottom right panel in Supplementary Fig. S34).

The second approach is to manually register the two modalities using the napari plugin imsmicrolink[10]. Essentially, the process is to first load the MSI data and PostMSI microscopy image and then manually select pixels and ablation marks that correspond to each other. Preferably, the PostMSI microscopy is registered to the MSI data to avoid potential resampling artifacts, but the other way around (i.e., register MSI-to-PostMSI) is also possible when the MSI was measured in multiple steps or when the rotation or shear is small.

Co-registration of all microscopy images is done using wsireg[49], which itself relies on elastix[50,51]. The registration steps are done as shown in box "Coregistration" in Fig. 1b. The microscopy image after IMC measurement (PostIMC) is registered with the microscopy image before measurement of IMC (PreIMC) using an affine transformation. A linear transformation is possible here because the two images are from the same tissue slice with the only difference being the ablation marks caused by the IMC acquisition. The registration between PreIMC and the microscopy image before MSI measurement (PreMSI) is done in two steps, first an affine transformation and then a non-linear transformation (b-splines) since the samples stem from two different (although similar because adjacent) tissue slices. Usually the registration can be done using the whole tissue slides, but in cases where the discrepancy in the position of the TMA cores between the two slides is high, it is also possible to manually select regions (i.e., samples) that correspond to each other and specifically register those two regions to each other. The selection of regions can be done in QuPath[52]. The registration between PreMSI and PostMSI can then again be done using an affine transformation. Because PreMSI and PostMSI images are visually quite different it is sometimes also necessary to select regions (i.e., TMA core locations) to be considered for registration. Again, the regions can be selected using QuPath.

For the registration of IMC to the microscopy images two possibilities exist. The first is by automatically detecting the ablation area making use of the known shape and area of the IMC image. After binarization registration with SimpleITK (function `Euler2DTransform`) is performed. Followed by optional fine-tuning with coherent point drift[47,48] using points at the boundary of both IMC and ablation mark. The second option is to manually select the ablated area on the tissue from the IMC acquisition in the PostIMC image which define the location of the IMC data. This can be done in QuPath by choosing regions and then exporting for registration. The IMC is then simply transformed using a rigid or affine transformation to fit within the selected region.

Now all transformations from IMC to MSI are known and the IMC mask images (from the above-described IMC workflow) can be transformed. Now the overlap of each cell with each MSI pixel can be calculated. This is done by a set of geometric operations by first creating polygons of the cells and MSI pixels and then calculating the intersection areas. Furthermore the cell centroids and areas are measured and included for downstream analysis and visualization.

## Co-registration: Precision of registration

For the evaluation of the precision of registration different metrics are calculated. The first is the Median Landmark Distance (MLD) which is defined as follows

$$MLD = \begin{cases} \| s_{m+1} - t_{m+1} \| & \text{if } n = 2m+1 \\ \frac{1}{2}(\| s_m - t_m \| + \| s_{m+1} - t_{m+1} \|) & \text{if } n = 2m \end{cases} \quad (3)$$

for $d_i = \|s_i - t_i\|$ the euclidean norm between landmark $s_i$ from the first image and landmark $t_i$ from the second image for all $(d_1, d_2, \ldots, d_n)$ with $d_1 < d_2, d_2 < d_3, \ldots, d_{n-1} < d_n$. The second metric that is calculated is the DICE coefficient $D$ with

$$D(S, T) = 2 \frac{|S \cap T|}{|S| + |T|}$$

between two binary masks $S$ and $T$. The extraction of both the landmarks and the masks depends on the image modality and is described in the following.

To evaluate the precision of the registration between the MSI image and PostMSI microscopy image the MLD of the automatically detected ablation marks and the corresponding MSI pixel after transformation are calculated.

To evaluate the precision of the registration between the microscopy images (PostMSI, PreMSI, PreIMC, PostIMC) three metrics are calculated.

The first metric is the MLD as defined above using automatically detected landmarks. The landmarks are extracted using the following procedure. First, for each TMA core RoMA[53] with the pretrained 'outdoor' model is applied for matching followed by sampling of matching points. Next, inliers are detected using a version of RANSAC[54] (sampling with PROSAC[55], scoring with MAGSAC[56]). Alternatively, the following procedure can be done. For each microscopy image the image is cut to the position of the expanded (by $200\mu m$ in all directions) IMC location. The resulting image is then first converted to gray scale and then multiple smaller, partially overlapping images are created where feature extraction and feature descriptors using KAZE[57] are calculated. The feature matching is done in multiple steps. The first is filtering by physical distance using a threshold of $50\mu m$. The second is to apply a nearest neighbor distance ratio thresholding in the feature descriptor space (lowest distance to second lowest distance with everything lower than 0.8 is considered a match) similar to the one described in ref. [58]. Next, one to many matches are removed, keeping only the pair of matches with the lowest distance in feature descriptor space. Then points on the same image are required to have no neighboring points closer then $10\,\mu m$. Lastly inliers are detected using a version of RANSAC[54] (sampling with PROSAC[55], scoring with MAGSAC[56]).

The second metric is the MLD between the centroids of the tissue masks that are extracted using the python packages rembg[59–61] and segment-anything[62]. The third metric are the MLDs and DICE coefficients between matching regions of the tissue (e.g., blood vessels). These regions are extracted using segment anything[62] with thresholds for minimum ($36^2 um^2$) and maximum ($512^2 um^2$) area. Matching of regions is done using multiple criteria. The first is based on area ($|log_{10}\left(\frac{area(A)}{area(B)}\right)| < log_{10}(1.5)$, with $A, B$ two regions). The second is based on the sum of differences of seven invariant Hu moments[63]:

$$I(A, B) = \sum_{i=1\ldots7} \left| m_i^A - m_i^B \right|$$

with $m_i$ the Hu moments. Regions that have similar area (as defined above) and minimal $I$ are assumed to match. Furthermore matches are filtered by maximum distance between centroids ($50um$) and if they overlap (DICE coefficient $> 0$).

Directly extracting landmarks and regions for the PostMSI image is difficult because of the ablation marks. Therefore, another preprocessing step is done for the PostMSI image to reduce the visibility of the ablation

marks. The procedure goes as follows. First, the image is transformed to frequency and magnitude space using the discreet Fourier transform. Next, regular grid location with high magnitude are detected by thresholding on the difference between magnitude and local threshold with the threshold being the 99% quantile. Furthermore, the center of the image is defined as being below the threshold. The obtained binary mask is then first convolved with a median filter to remove single pixels and then dilated to get more round areas. The magnitudes of detected locations is replaced with a local mean of the magnitudes. Finally the inverse discreet Fourier transform is applied.

For the evaluation of the precision of registration of IMC-to-PostIMC the position of the IMC ablation mark is compared to the IMC location. For that, first the IMC ablation mark on the PostIMC is detected by thresholding in the saturation channel in the HSV color space. Next an inverse distance transform image (0 if no mask, >0 if mask with max at the boundary) is calculated for both the IMC location and IMC ablation mask. Then the resulting images are co-registered using an Euler transformation. Lastly, a set of regularly spaced point are generated on the IMC ablation mask, transformed using the previously estimated transform and distances before and after transformation are calculated.

## Co-registration: MSI-to-PostMSI Manual vs. Automatic registration

To investigate the precision of the automatic registration of the MSI to the PostMSI microscopy image multiple samples were randomly chosen for manual registration to compare against the automatic registration. The median distance between MSI pixels (after registration) and MSI ablation marks was calculated for the different samples. For three samples, the manual registration was performed three times each to investigate consistency.

## Co-registration: Microscopy Manual vs. Automatic registration

In SCiLS Lab manual registration of a 40x high resolution H&E stained image to a low resolution 2400 dpi slide scan (after MALDI Matrix application) was performed according to instructions from SCiLS Lab 2023a (manually selecting two landmarks on the two modalities that correspond to each other). To evaluate the precision of this registration 1–3 landmarks were manually selected per TMA core (leading to 64 landmarks in total). The euclidean distances between those landmarks are then compared. Furthermore, automatic registration based on the manual co-registration was performed (using wsireg in the same way as described above) and the same landmarks were used to calculate distances. Additionally, to compare to the workflow presented here the distances of matching landmarks as described in Section 'Precision of registration' were calculated between PostIMC and PreIMC images.

## Co-registration: IMC-to-PostIMC

To evaluate the precision of the IMC and the PostIMC microscopy image an additional experiment was performed. For this study, three needle biopsy samples of human liver were dewaxed and then subjected to staining the 191/193 Ir DNA Intercalator. A total of 10 ROIs were then acquired on IMC with a low laser power of −4 dBa, followed by an H&E stain was performed on the same tissue. The nuclei were then detected using stardist[64,65] with the corresponding pretrained model for nuclei detection in H&E stained images. Since after the acquisition of IMC the remaining tissue is slightly different (effectively resulting in data from two adjacent regions) the number of cells and their area changes. Therefore only cells that overlap in the two modalities are used to calculate the DICE coefficient and MLD of the centroids.

## Per sample aggregated testing for associations between IMC and MSI

The statistical model used to test for the association of a m/z value (continuous response) with the cell type proportion (compositional covariate) follows the model described in ref. 66. In short, the cell type proportions are transformed to pivot coordinates and the least squares method is used for estimation. Orthogonal transformations between pivot coordinates respective to the individual cell type proportions are employed and estimation is repeated each time resulting in the final estimates. Additional non-compositional covariates include condition and TMA.

## Testing for associations between IMC and MSI

Testing for associations between cell types and m/z values is done in two steps.

The first step is to fit for each sample and each m/z value a Simultaneous Autoregressive Model (SAR) with the following specification (using spatialreg[67]).

$$\mathbf{y} = \rho \mathbf{W} \mathbf{y} + \mathbf{X}\beta + \epsilon \qquad (4)$$

For the total number of MSI pixels $n$ and the total number of cell types $c$, $\mathbf{y} = \{y_1, \ldots, y_n\}$ are the TIC normalised m/z values, $\rho$ the parameter for the strength of spatial dependency between neighbouring pixels, $\mathbf{W} \in \mathbb{R}^{n \times n}$ the (row-standardized) spatial weight matrix (>0 if neighbour, =0 else), $\mathbf{X} \in \mathbb{R}^{n \times (c+1)}$ the areas per cell type in an MSI pixel (normalised by the total area of a MSI pixel), $\beta = \{\beta_0, \ldots, \beta_c\}$ the parameters of interest, $\epsilon_i$ the error ($\mathcal{N}(0, \sigma^2)$). Parameters $\beta$ will then be used in the second fitting step. Alternatively, instead of using the cell area proportions in $\mathbf{X}$ IMC markers aggregated per MSI pixel can be used.

The second step is to fit a (mixed) linear regression model for each m/z value using the previously estimated coefficients $\beta$ as response and sample specific metadata ($\mathbf{Z}$) as covariates.

$$\beta = \mathbf{Z}\gamma + \epsilon \qquad (5)$$

where $\gamma$ are the coefficients to be estimated and $\epsilon$ the errors. Additionally the inverse of the variances of the $\beta$ are included as weights in the fitting process.

## Simulation

The overview of the Simulation is given in Supplementary Fig. S13. The simulated images have a size of 1000 um x 1000 um in which a multi type point pattern with simple sequential inhibition (inhibition distance = 0.975 quantile sqrt(area/pi) of all observed cell areas) is sampled. The intensity is inhomogeneous based on a sampled Gaussian random field (GRF, scaled from 0 to 1) where the number of cells is fixed for each simulation condition. Multiple celltypes are simulated, the first has intensity according to the simulated scaled GRF while all others have intensity inverse to the GRF. Cell areas are simulated from a negative binomial with the parameters obtained from fitting on the real cell areas. The distribution of m/z values can be described with a log-normal distribution. Therefore, to model the m/z value of a cell (per $\mu m$ squared) a log-normal distribution is chosen (parameters manually specified). Those parameters can be adjusted per celltype and are the main parameters to evaluate the sensitivity and power.

Next a grid is overlaid representing the MSI pixels. The stepsize is the same as in the experimental data which is 30 $\mu m$ with a pixelsize of 24 $\mu m$. Other parameters are stepsize of 20 and 10, and pixelsize of 30,24,20,10,8. To investigate how important exact registration is a translation is applied in the x-axis of either, 0, 1, 5 or 10 $\mu m$. Furthermore the effect of imprecise cell areas are modeled. First by randomly reshuffling of the cell areas (extreme case) or by a rank based reordering (split the areas into multiple subsets based on quantiles, within each subset invert order of ranks of areas). This second approach tries to mimic the fact that IMC and MSI were not obtained from the same tissue slice but from adjacent slices, meaning the cell areas will not be exactly the same. Randomly adding or removing cells would have been another option which wasn't used here. For those last three parameters the estimates from model fitting can be directly compared to each other since they rely on the same underlying data. Next, three different error terms are added to the aggregated MSI image. The first is an error based on the neighbouring pixels, essentially smoothing the signal. The second is an uncorrelated error sampled from a log-normal. The third is

correlated error drawn from a GRF scaled between 0 and a max value (of 500). Additionally the cell type areas per pixel are aggregated as well leading to the final structure of the observed data.

## Reanalysis Nunes et al. data

Publicly available data was retrieved from https://figshare.com/s/c58ddc70fa8dc0602842, consisting of processed MSI images, processed and annotated IMC images. Automatic registration between IMC and MSI was done using SimpleITK (function `AffineTransform` with metric to optimize set as mean squares)[46] with three similar MSI channels (0002, 0003, 0004) that were aggregated and one similar corresponding IMC channel (Keratin). Both Euler and Affine transformation models were optimized. The evaluation of distances between the manual and automatic registrations consisted on transforming a set of evenly spaced points with the three available transforms and then calculating the differences between the manual vs euler and manual vs affine transformed points. The data was aggregated both to the single cell level (for plotting) and the MSI pixel level (for statistical modeling). The association testing was done as described above.

## Artificial Tissue - analysis

A single slide with 5 spots (3 with single cell lines, 2 with mixture) was prepared and analyzed as described in the following. Selection of m/z values was done as described in the following. First, a list of candidate values was obtained from Metaspace using the SwissLipids and HMDB databases[45]. Then the single cell line samples were compared to each other to find differentially 'expressed' peaks. For that, first the individual MSI measurement were filtered to only contain measurements with actual cells present. Additionally measurements with low total ion currents where removed as well. Additionally m/z were filtered to exclude m/z values that showed higher intensity close to regions with no cells. Then t-tests (using 'scran'[43]) were run for the candidate peaks across samples. Highest difference peaks that where also significantly positive were selected and underwent an additional round of visual inspection which resulted in the final peak list. IMC data processing was done as described above with the difference that cell type annotation was done based on the intensity of the two markers for cell lines ac16 and mel624 resulting in four different clusters: ac16+mel624-, ac16-mel624+, ac16-mel624-, ac16+mel624+. Statistical modeling was done as described above for testing lipid intensity vs. cell type proportion, additionally for testing lipid intensity vs. cell marker intensity the average cell marker intensity per MSI pixel was first calculated and then used in the modeling as the covariates.

## Case Study - analysis

Two TMAs with a total of 68 cores were prepared and analyzed as described in the following. Processing of IMC, MSI, Co-registration, and statistical modeling was done as described above. Additionally, domains (Hepatocytes, Inflammation) were defined on the IMC markers HepPar1 and Collagen-1 at the single cell level. First, for each cell an weighted intensity based on neighboring cells with a distance of maximum 40 microns was constructed. Second, the two markers were scaled by dividing by the 0.99 quantile. Next, the initial cluster was assigned based on which marker had higher intensity. This was followed by a smoothing by considering the cluster labels of the neighboring cells and change of the cluster label based on the majority. At the MSI pixel level, each MSI pixel was assigned the domain with the maximum number of single cell labels. If no cells were present the label of the closest labeled pixel was used instead. For the statistical testing of domain vs. m/z value, first the average m/z values for each sample and domain were calculated. Second a linear model using the m/z values as response and the domain (as categorical variable) as covariate. Additionally the slide identifier was used as another covariate (m/z = domain + TMA_type).

## Reporting summary

Further information on research design is available in the Nature Portfolio Reporting Summary linked to this article.

## Data availability

Raw IMC and MALDI-MSI data are deposited at the BioImage Archive under accession number: S-BIAD2091. Processed tabular data is available on Zenodo via: https://doi.org/10.5281/zenodo.15641167.

## Code availability

MIMIC is available at: https://github.com/retogerber/MIMIC (Zenodo: https://zenodo.org/records/18780816). The IMC processing pipeline is available at: https://github.com/retogerber/imc_workflow (Zenodo: https://zenodo.org/records/18780899). Analysis scripts are at: https://gitlab.uzh.ch/retogerber/imc_to_ims_methods_analysis_scripts (Zenodo: https://zenodo.org/records/18780973).

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

## Acknowledgements

We thank Reinhard Furrer for help with the statistical model, Heath Patterson for help with the co-registration and all members of the Robinson and Guglietta Lab for helpful feedback. We thank the MUSC proteomics center for their help with MALDI imaging. This publication was supported, in part, by the National Center for Advancing Translational Sciences of the National Institutes of Health under Grant Numbers TL1 TR001451 & UL1 TR001450. Additional support by the MUSC Vice President for Research, College of Medicine, Hollings Cancer Center (P30 CA138313), MUSC Digestive Disease Research Core Center (P30 DK123704), Medical Scientist Training Program (T32 GM008716), and R01 CA258882. Supported in part by the Flow Cytometry and Cell Sorting Shared Resource, Hollings Cancer Center, Medical University of South Carolina (P30 CA138313). MDR acknowledges support from the University Research Priority Program Evolution in Action at the University of Zurich, as well as the Swiss National Science Foundation (project grant 310030_204869).

## Author contributions

Data acquisition: J.G., C.K.; Data analysis: R.G. and M.D.R.; Conceptualization and data interpretation: R.G., J.G., C.K., M.D.R., and S.G.; Funding: C.K., M.D.R., S.G.; Writing: S.G., C.K., M.D.R., R.G., J.G.

## Competing interests

The authors declare no competing interests.
