## [Transparent Peer Review file · Communications Biology]

MIMIC: a flexible pipeline to register and summarize IMC-MSI experiments

Corresponding Author: Professor Mark Robinson

This manuscript has been previously submitted at another journal. This document only contains information relating to versions considered at Communications Biology.

Version 0:

Reviewer comments:

Reviewer #1

(Remarks to the Author)

I'd like to extend my apologies for the slight delay in this review, as the start of the semester has been quite demanding. With that said, it was a genuine pleasure to review this manuscript. The authors present a truly impressive piece of work, and their MIMIC pipeline stands to be a tremendously useful tool for the spatial biology community.

This paper introduces MIMIC, a computational pipeline designed to integrate data from Mass Spectrometry Imaging and Imaging Mass Cytometry. The primary challenge addressed is the co-registration of these modalities, which have different resolutions and data characteristics. MIMIC provides a semi-automated workflow that uses a series of microscopy images as a scaffold to align the data accurately. The pipeline includes rigorous quality control steps, such as registration error assessment, and employs pixel-level statistical models to identify associations between analytes and cell types. The authors validate their approach using a synthetic tissue with a known ground truth and a re-analysis of a public dataset, demonstrating improved registration and more robust results. They further apply MIMIC to a study of human liver tissue, successfully identifying known and novel associations between glycans and specific cell types in the context of metabolic dysfunction-associated steatotic liver disease.

The work is well-executed and clearly presented. I have just a few suggestions that I believe could further strengthen the manuscript.

1, To further contextualize the work and highlight its novelty, it might be beneficial to provide a more comprehensive discussion of MIMIC's place within the current landscape of co-registration tools. While the review of previous work is thorough a comparative table summarizing the features of MIMIC versus other pipelines (supported modalities, statistical methods, etc.) would offer a clear, at-a-glance view of its specific advantages. Additionally, discussing very recent alternative strategies for bridging omics layers, such as the approach detailed in <https://www.biorxiv.org/content/10.1101/2025.02.23.639721v1>, would provide valuable contemporary context.

2, to maximize the impact and utility for the broader research community, have the authors considered releasing the generated data in a format compatible with standard spatial biology analysis packages? Providing the data in a format like AnnData (H5ad) for example, would greatly facilitate re-analysis and integration with other datasets and promote wider adoption of the MIMIC pipeline's outputs.

3, The authors are to be commended for their direct approach to addressing registration error, including its incorporation into simulation studies. The practical implications of these errors on the final biological conclusions could be explored a bit more deeply to further bolster the findings. A sensitivity analysis, for instance, showing how varying levels of registration error effects the detection of analyte-cell type associations would be a powerful addition.

Reviewer #2

(Remarks to the Author)

The authors developed a pipeline to co-register and integrate MSI and IMC data. Even though parts of the workflow have been inspired by previous works which are correctly referenced by the authors, to the best of this reviewer's knowledge, this work is novel in the way it proposes the use of microscopic images to accurately align multimodal data. This is of significant value for the scientific community, since it highlights how the acquisition of a high-resolution and widely accessible modality such as microscopy can significantly improve and simplify multimodal co-registration. Also, microscopy without staining can be acquired alongside many, if not all, spatial/imaging technologies, making this an interesting approach for a wide range of technologies.

To this reviewer, the work looks original, convincing, and I believe it has the potential to influence thinking in the field. The statistical analysis looks appropriate and valid. I cannot fully judge on the ability of a researcher to reproduce the work, as I didn't try to reproduce the results myself. However, given the detailed description of the methods and the availability of the code and the data, I believe that this should be possible.

Given what was just stated, I have three major questions/comments, and several minor questions/comments.

Major Questions

1) Methods - Co-registration. . "First the PostMSI microscopy image is segmented to obtain the mask of the tissue samples (TMA cores). Next the MSI regions (i.e., connected components) are mapped to the corresponding tissues on the PostMSI microscopy image by registering the binary masks using SimpleITK [44] (top panel in Supplementary Figure S27)." How much is the alignment of MSI data and PostMSI images dependent on the shape of the MSI data? In most cases the MSI operator selects a region of interest that has a very similar shape to the tissue section. This enables the alignment of the MSI data to the tissue by aligning the edges of the tissue section. In fewer cases, especially when a tissue section is very big, the MSI operator chooses a region that is smaller and included inside the tissue. Would you, in this case, try to make a mask only of the ablated region, rather than of the tissue section, as you are doing for the IMC data?

2) "Both aggregation approaches, at the sample- or domain-level, potentially average out useful information that can be retained only if directly looking at the pixel-level values. For instance, m/z 2067.684 clearly correlated with hepatocytes at the pixel-level for a single sample (Figure 6a-d) and also across all samples (Figure 6e), but did not at the bulk (aggregated per sample) or pseudobulk (aggregated per sample-domain) level (Supplementary Figure S22, S23)." The manuscript contains data from samples with quite low spatial heterogeneity, especially slides seeded with a densely packed layer of single cells or liver tissue sections. This could make MIMIC's task of aligning the data harder, but at the same time, a not-so-perfect alignment could be missed when it comes to the results of downstream analysis (lipids or glycans-to-cell-type association, or finding differences between compartments in the liver). I wonder if a better showcase would be to use tissues that have more distinct anatomic layers or regions such as mouse brain or skin. However, I don't intend to slow down the review process with this comment, so I leave it to the authors to decide if adding an analysis of such data is easily achievable (even from public available sources), or just replying to this comment with their opinion in the rebuttal letter.

3) Discussion - "Furthermore, since the registration workflow is structured around the intermediate microscopy images, extending MIMIC to include other modalities, such as spatial transcriptomics, is possible if the respective modality can itself be registered to a microscopy image." I agree with these statements. However, some other modalities in their commercial versions are currently aligned to microscopy images using landmark-based alignments. This is valid, for example, for Bruker's MSI solution (whose landmark-based with two-point approach is used also in this work) and 10x Genomics' Visium and Xenium. My question is: is the possibility to align other data types to a microscopy image really the only requirement that is needed to extend MIMIC to other modalities? Or do you think that the alignment of these other modalities to microscopy images should be done with a non-landmark-based approach, such as the one based on ablation marks detection that you propose for IMC and MSI data? I think it could be beneficial for the readers if the authors could more precisely detail the requirements to extend MIMIC to other modalities, if they see fit.

Minor questions/comments:

4) The registration performance has been evaluated on liver tissue sections using automatically detected landmarks, and on the artificial tissues created with single cell using the correlation between a biological feature (lipids detection) and the cell types expected to contain that feature. Would such a 'biological readout approach' be possible also on the liver tissue sections and provide an equally/comparably robust benchmark of the registration performance?

5) "The microscopy image after IMC measurement (PostIMC) is registered with the microscopy image before measurement of IMC (PreIMC) using an affine transformation. A linear transformation is possible here because the two images are from the same tissue slice with the only difference being the ablation marks caused by the IMC acquisition. The registration between PreIMC and the microscopy image before MSI measurement (PreMSI) is done in two steps, first an affine transformation and then a non-linear transformation (b-splines) since the samples stem from two different (although similar because adjacent) tissue slices". Different microscopy images could be acquired on slightly different planes which might not even be exactly parallel. Moreover, as any technique, microscopy is prone to acquisition errors. Finally, many experimental scientists believe that the tissue section can be altered during the analysis (MSI, IMC, or other technique), and cause changes in the tissue dimensions, position, or damage the tissue section. Do you think that registering also the pre- and post-technique images with a non-linear approach might be beneficial? Also, have you investigated if such alterations (changing of tissue dimensions/position, damage) actually happen in the data that you have analyzed? If so, do they have a significant impact on the registration performance, and how should the researcher deal with them?

6) The comparison with a high-resolution H&E registered to a low-resolution scanned image gave pretty high MLD scores. Would you discourage this kind of practice? Do you think there could be a better practice that can be proposed to the readers?"

7) Introduction - "Complementing these are mass spectrometry imaging (MSI) techniques, which capture spatio-temporal changes in glycans, lipids, metabolites, and even drugs". MSI can be used to detect proteins well, as accurately referenced with reference [4] just in the following sentence. Even though the sentence is correct, some readers might think that MSI cannot be used to detect proteins because of the use of the word 'complementing' and because proteomics is not included in the list. I suggest including proteomics in the list and maybe rephrasing to avoid misunderstandings.

8) Introduction - "The same holds for SpaceM, which additionally cannot co-register adjacent slices, but on the other hand has automated MSI ablation detection [11]". This sentence is correct. However, since SpaceM was mainly developed to enable single-cell metabolomics, I suggest rephrasing so that it becomes clear that the automatic MSI ablation tool might be used also on tissue sections, but the co-registration of adjacent sections is not part of the method, probably because tissue sections were not the main goal of the technology.

9) Introduction - "Therefore, solving the "pixel gap" when combining MALDI-MSI (which does not reach single cell resolution) with single cell based technologies remains a challenge." Since there are cells that are 10 μm big or more, I believe that 5 μm resolution MSI can be considered single-cell resolution, although I agree that a 'pixel gap' still needs to be solved. I believe that MALDI-MSI not being able to reach single-cell resolution is a debatable statement, even though in real-life scenarios it feels about true. My suggestion would be to rephrase this sentence to avoid misunderstandings, or to better justify why MALDI-MSI cannot reach single-cell resolution.

10) "Furthermore, as expected, manual registration resulted in small inconsistencies across repeated registration compared to an automatic registration." Do you mean that the inconsistencies were smaller for manual registration than for automatic registration? Or that automatic registration resulted in smaller inconsistencies than manual registration? Please rephrase to improve clarity.

11) "cell level MSI intensity differences between manual and Euler registration, log₂-fold change of example MSI channel of manual vs. Euler registration with the cell area on the x-axis. Boxes show the percentage of cells with a log₂-fold change higher than log₂(1.5), lower than log₂(1.5) or in between." What do the dashed lines represent in this figure? Can you indicate this in the caption? It seems they roughly correspond to log₂(0.5) and -log₂(0.5): did you mean these values in the caption instead of log₂(1.5) and -log₂(1.5)?

12) "Quantitatively, the observed MSI-pixel-level associations between m/z 728.52 and MeI624 are shown in Figure 4c". Is it possible to quantify this association with a number in the same image? Maybe a correlation coefficient? Would it be the same scaled coefficient of association shown in Figure 4d?

13) Reanalysis Nunes et al. data - I believe that this paper didn't publish all the data you ideally need for your workflow (e.g. PostMSI ablation marks). Do you think this affected the performance of MIMIC (or probably cannot even be called MIMIC since not all required data were available)? If so, did it affect it significantly? Do you think that the performance of MIMIC on this data could be improved if all the ideal data were available?"

14) Results - "While the estimates of association do not seem to depend on the abundance of a cell type (top annotation and left heatmap in Figure 6f), the -log₁₀ FDR corrected p-values are, as expected, on average negatively correlated with the proportion of missing m/z values or in other words, positively correlated with number of observations (Supplementary Figure S25". When you write '(top annotation and left heatmap in Figure 6f)' do you mean 'top and left barplot in Figure 6f'?

Version 1:

Reviewer comments:

Reviewer #1

(Remarks to the Author)

Congratulations to the authors for such a good work.

Reviewer #2

(Remarks to the Author)

I am fully satisfied with the answers provided by the authors in their rebuttal letter and with the modifications they have applied to the text and images.

Throughout the response to reviewers, we quote the reviewers in *blue italics* and respond in non-italic black font. Some of the next text that we have written in the manuscript is shown in non-italic red font.

Referee expertise:

Referee #1: Deep learning, Spatial Omics

Referee #2: Mass Spectrometry Imaging and Imaging Mass Cytometry

Reviewer 1

I'd like to extend my apologies for the slight delay in this review, as the start of the semester has been quite demanding. With that said, it was a genuine pleasure to review this manuscript. The authors present a truly impressive piece of work, and their MIMIC pipeline stands to be a tremendously useful tool for the spatial biology community.

This paper introduces MIMIC, a computational pipeline designed to integrate data from Mass Spectrometry Imaging and Imaging Mass Cytometry. The primary challenge addressed is the co-registration of these modalities, which have different resolutions and data characteristics. MIMIC provides a semi-automated workflow that uses a series of microscopy images as a scaffold to align the data accurately. The pipeline includes rigorous quality control steps, such as registration error assessment, and employs pixel-level statistical models to identify associations between analytes and cell types. The authors validate their approach using a synthetic tissue with a known ground truth and a re-analysis of a public dataset, demonstrating improved registration and more robust results. They further apply MIMIC to a study of human liver tissue, successfully identifying known and novel associations between glycans and specific cell types in the context of metabolic dysfunction-associated steatotic liver disease.

The work is well-executed and clearly presented. I have just a few suggestions that I believe could further strengthen the manuscript.

1, To further contextualize the work and highlight its novelty, it might be beneficial to provide a more comprehensive discussion of MIMIC's place within the current landscape of co-registration tools. While the review of previous work is thorough a comparative table summarizing the features of MIMIC versus other pipelines (supported modalities, statistical methods, etc.) would offer a clear, at-a-glance view of its specific advantages. Additionally, discussing very recent alternative strategies for bridging omics layers, such as the approach detailed in <https://www.biorxiv.org/content/10.1101/2025.02.23.639721v1>, would provide valuable contemporary context.

Thank you for this relevant comment. We added a Supplementary Table to compare the described computational tools and allow for easier comparison with MIMIC. Additionally, we added a small section referring to possible downstream analysis.

More specifically, we have added the following:

After successful co-registration, downstream analysis can involve among others statistical testing of associations, lower dimensional projection via factor analysis, or prediction of single-cell omics.

2, to maximize the impact and utility for the broader research community, have the authors considered releasing the generated data in a format compatible with standard spatial biology analysis packages? Providing the data in a format like AnnData (H5ad) for example, would greatly facilitate re-analysis and integration with other datasets and promote wider adoption of the MIMIC pipeline's outputs.

The output of MIMIC is a set of CSV files, which allows widespread, cross-platform use. For the analyses conducted in the manuscript, such files are already made (publicly) available on zenodo. Additionally, co-registered ome.tiff files are generated.

3, The authors are to be commended for their direct approach to addressing registration error, including its incorporation into simulation studies. The practical implications of these errors on the final biological conclusions could be explored a bit more deeply to further bolster the findings. A sensitivity analysis, for instance, showing how varying levels of registration error affects the detection of analyte-cell type associations would be a powerful addition.

We included a sensitivity analysis for both the artificial tissue as well as the case study and updated the manuscript.

Text is added in two separate places:

A sensitivity analysis (i.e., increasing registration errors), shows a convergence of the estimates towards zero (Supplementary Figures S18-S19).

A sensitivity analysis reveals that estimates tend to converge towards zero for larger registration errors for cell types with large local spatial heterogeneity (e.g., CD8 cells), while for cell types with smaller local spatial heterogeneity (e.g., Hepatocytes), the estimates are more stable (Supplementary Figure S28).

Reviewer 2

The authors developed a pipeline to co-register and integrate MSI and IMC data. Even though parts of the workflow have been inspired by previous works which are correctly referenced by the authors, to the best of this reviewer's knowledge, this work is novel in the way it proposes the use of microscopic images to accurately align multimodal data. This is of significant value for the scientific community, since it highlights how the acquisition of a high-resolution and widely accessible modality such as microscopy can significantly improve and simplify multimodal co-registration. Also, microscopy without staining can be acquired alongside many, if not all, spatial/imaging technologies, making this an interesting approach for a wide range of technologies.

To this reviewer, the work looks original, convincing, and I believe it has the potential to influence thinking in the field. The statistical analysis looks appropriate and valid. I cannot fully judge on the ability of a researcher to reproduce the work, as I didn't try to reproduce the results myself. However, given the detailed description of the methods and the availability of the code and the data, I believe that this should be possible.

Given what was just stated, I have three major questions/comments, and several minor questions/comments.

Major Questions

1) Methods - Co-registration. . "First the PostMSI microscopy image is segmented to obtain the mask of the tissue samples (TMA cores). Next the MSI regions (i.e., connected components) are mapped to the corresponding tissues on the PostMSI microscopy image by registering the binary masks using SimpleITK [44] (top panel in Supplementary Figure S27)." How much is the alignment of MSI data and PostMSI images dependent on the shape of the MSI data? In most cases the MSI operator selects a region of interest that has a very similar shape to the tissue section. This enables the alignment of the MSI data to the tissue by aligning the edges of the tissue section. In fewer cases, especially when a tissue section is very big, the MSI operator chooses a region that is smaller and included inside the tissue. Would you, in this case, try to make a mask only of the ablated region, rather than of the tissue section, as you are doing for the IMC data?

We thank the reviewer for this question. This stage of the analysis is simply to match TMA cores and MSI regions and no high precision of co-registration is needed, only the binary masks (of TMA core and MSI region) need to overlap after registration. The shape should not have a big influence on this step. However, there should not be any holes in the ablated region. Also in the artificial tissue experiment, we have exactly this setting with the MSI region smaller than the actual "tissue" extent. The actual registration per core happens then in the next steps.

We updated the method section to clarify this step of the analysis:

This step is required to subsequently register individual cores, the co-registration precision is not critical at this stage. Therefore, the shape of the MSI regions does not need to match the shape of the TMA cores, but the MSI region must be completely filled.

2) "Both aggregation approaches, at the sample- or domain-level, potentially average out useful information that can be retained only if directly looking at the pixel-level values. For instance, m/z 2067.684 clearly correlated with hepatocytes at the pixel-level for a single sample (Figure 6a-d) and also across all samples (Figure 6e), but did not at the bulk (aggregated per sample) or pseudobulk (aggregated per sample-domain) level (Supplementary Figure S22, S23)." The manuscript contains data from samples with quite low spatial heterogeneity, especially slides seeded with a densely packed layer of single cells or liver tissue sections. This could make MIMIC's task of aligning the data harder, but at the same time, a not-so-perfect alignment could be missed when it comes to the results of downstream analysis (lipids or glycans-to-cell-type

association, or finding differences between compartments in the liver). I wonder if a better showcase would be to use tissues that have more distinct anatomic layers or regions such as mouse brain or skin. However, I don't intend to slow down the review process with this comment, so I leave it to the authors to decide if adding an analysis of such data is easily achievable (even from public available sources), or just replying to this comment with their opinion in the rebuttal letter.

We thank the reviewer for highlighting our dilemma. Both the co-registration of MSI to PostMSI as well as IMC to PostIMC are agnostic to both measured intensities (of MSI or IMC), as well as the spatial structure of the tissue. The only intensity-based requirement for registration is that the ablation marks are visible on the microscopy images. However, the microscopy image co-registration can be more challenging if distinct spatial features are missing. In our setting, working with TMA cores, however this shortcoming is at least partially mitigated because of the global shape of a single core (i.e., round).

Our main research interest lies in investigating glycans in the liver, and thus we currently don't have data from other tissues with other MSI analytes, and we were not able to find publicly available datasets that contains all the necessary data required to run MIMIC (MSI, IMC, microscopy images). However, as mentioned to Reviewer 1, we added a small sensitivity analysis to show that the estimates of the association between celltypes and MSI analytes in general tend to converge towards zero for larger co-registration errors.

3) Discussion - "Furthermore, since the registration workflow is structured around the intermediate microscopy images, extending MIMIC to include other modalities, such as spatial transcriptomics, is possible if the respective modality can itself be registered to a microscopy image." I agree with these statements. However, some other modalities in their commercial versions are currently aligned to microscopy images using landmark-based alignments. This is valid, for example, for Bruker's MSI solution (whose landmark-based with two-point approach is used also in this work) and 10x Genomics' Visium and Xenium. My question is: is the possibility to align other data types to a microscopy image really the only requirement that is needed to extend MIMIC to other modalities? Or do you think that the alignment of these other modalities to microscopy images should be done with a non-landmark-based approach, such as the one based on ablation marks detection that you propose for IMC and MSI data? I think it could be beneficial for the readers if the authors could more precisely detail the requirements to extend MIMIC to other modalities, if they see fit.

After running MIMIC co-registration, both MSI and IMC are co-registered to the microscopy images. Therefore, in principle, the ability to co-register to the same microscopy images is really the only requirement to extend MIMIC. However, this might pose some restrictions on the needed microscopy image itself, such as similar resolution. Ideally, all registration steps would be done automatically (for reproducibility) and include co-registration error metrics to ensure robustness.

We clarified this now in the manuscript:

As such, for successful co-registration, there are some requirements on the microscopy image itself, such as similar resolution and similar visible spatial structures. Ideally, additional modalities would be co-registered automatically (for reproducibility) and evaluate the precision of co-registration to ensure robustness. Furthermore, if modalities are acquired over more than two slides, the target microscopy image needs to be carefully chosen to reduce co-registration errors.

Minor questions/comments:

4) The registration performance has been evaluated on liver tissue sections using automatically detected landmarks, and on the artificial tissues created with single cell using the correlation between a biological feature (lipids detection) and the cell types expected to contain that feature. Would such a 'biological readout approach' be possible also on the liver tissue sections and provide an equally/comparably robust benchmark of the registration performance?

We thank the reviewer for highlighting yet another of our dilemmas: the lack of (a full) ground truth in our liver tissue experiment (or orthogonal public datasets with ground truth). While in the literature there are some associations between cell types and glycans described, which we also recover, there is no absolute ground truth available for our specific set of samples. Also, generating such a ground truth, e.g., mechanical dissociation of a liver and cell sorting, would likely disrupt glycan profiles. Therefore, we do not think it is possible with our liver tissue data to benchmark the performance of the co-registration using celltype-glycan associations.

5) "The microscopy image after IMC measurement (PostIMC) is registered with the microscopy image before measurement of IMC (PreIMC) using an affine transformation. A linear transformation is possible here because the two images are from the same tissue slice with the only difference being the ablation marks caused by the IMC acquisition. The registration between PreIMC and the microscopy image before MSI measurement (PreMSI) is done in two steps, first an affine transformation and then a non-linear transformation (b-splines) since the samples stem from two different (although similar because adjacent) tissue slices". Different microscopy images could be acquired on slightly different planes which might not even be exactly parallel. Moreover, as any technique, microscopy is prone to acquisition errors. Finally, many experimental scientists believe that the tissue section can be altered during the analysis (MSI, IMC, or other technique), and cause changes in the tissue dimensions, position, or damage the tissue section. Do you think that registering also the pre- and post-technique images with a non-linear approach might be beneficial? Also, have you investigated if such alterations (changing of tissue dimensions/position, damage) actually happen in the data that you have analyzed? If so, do they have a significant impact on the registration performance, and how should the researcher deal with them?

In our data, we observed slight vertical jumps in the microscopy images, possibly caused by vibrations during acquisition. These however were small (maybe around 1 micrometer) compared to general registration errors. We did not observe other acquisition artifacts such as slightly-different planes or possibly-altered samples between microscopy acquisitions. However,

for slightly different planes, the co-registration could be extended to a homography if an affine transformation is insufficient, but we are not aware that this is common practice. No large experimental alterations during analysis were observed (except the laser ablations). That is, no linear transformation did scale or shear the images (meaning an Euler transform should have worked fine as well for our data). But the large difference of PostMSI and PreMSI microscopy images (because of the MALDI matrix) means that small local changes to the tissue cannot be ruled out as they would be difficult to detect. A non-linear approach between PostIMC and PreIMS is almost certainly not necessary in most cases. Between PostMSI and PreMSI, a non-linear approach probably is also not needed or hard to achieve technically (because of the very different appearance).

We added a short sentence regarding these potential issues to the Discussion:

Furthermore, microscopy image acquisition artifacts, such as slightly-different planes or possibly-altered samples, might require image transformations with larger degrees of freedom for successful registration.

6) The comparison with a high-resolution H&E registered to a low-resolution scanned image gave pretty high MLD scores. Would you discourage this kind of practice? Do you think there could be a better practice that can be proposed to the readers?"

If precise co-registration is required, we indeed discourage the practice of using low resolution images for co-registration. Using high-resolution images and switching to automatic co-registration approaches seems like it could improve the quality of co-registration. The performance of automated histopathology co-registration methods has been evaluated and described in the literature previously (e.g., ACROBAT challenge).

We have added a reference to the reader to these resources, since we have not done a comprehensive benchmark of co-registration of microscopy images ourselves:

This suggests that high-resolution microscopy images and automatic co-registration (e.g., methods described in the ACROBAT challenge) can improve the quality of co-registration.

7) Introduction - "Complementing these are mass spectrometry imaging (MSI) techniques, which capture spatio-temporal changes in glycans, lipids, metabolites, and even drugs". MSI can be used to detect proteins well, as accurately referenced with reference [4] just in the following sentence. Even though the sentence is correct, some readers might think that MSI cannot be used to detect proteins because of the use of the word 'complementing' and because proteomics is not included in the list. I suggest including proteomics in the list and maybe rephrasing to avoid misunderstandings.

We have rephrased slightly and added proteins to the list:

In addition to these are mass spectrometry imaging (MSI) techniques, which capture spatio-temporal changes in glycans, lipids, metabolites, proteins, and even drugs.

8) Introduction - *"The same holds for SpaceM, which additionally cannot co-register adjacent slices, but on the other hand has automated MSI ablation detection [11]."* This sentence is correct. However, since SpaceM was mainly developed to enable single-cell metabolomics, I suggest rephrasing so that it becomes clear that the automatic MSI ablation tool might be used also on tissue sections, but the co-registration of adjacent sections is not part of the method, probably because tissue sections were not the main goal of the technology.

We have now clarified the intended setting of SpaceM and its capabilities in the manuscript:

The same holds for SpaceM, developed for single-cell metabolomics on cell cultures with MALDI-MSI plus light microscopy on the same slide, and therefore does not include capabilities to co-register adjacent slices, but on the other hand has automated MSI ablation detection, which could also be applicable on tissue sections.

9) Introduction - *"Therefore, solving the "pixel gap" when combining MALDI-MSI (which does not reach single cell resolution) with single cell based technologies remains a challenge."* Since there are cells that are 10 μm big or more, I believe that 5 μm resolution MSI can be considered single-cell resolution, although I agree that a 'pixel gap' still needs to be solved. I believe that MALDI-MSI not being able to reach single-cell resolution is a debatable statement, even though in real-life scenarios it feels about true. My suggestion would be to rephrase this sentence to avoid misunderstandings, or to better justify why MALDI-MSI cannot reach single-cell resolution.

We thank the reviewer for challenging this statement. We rephrased this part and removed the direct claim about MALDI-MSI reaching single-cell resolution or not:

Therefore, solving the "pixel gap" when combining MALDI-MSI (where reaching single cell resolution is currently challenging) with single cell based technologies remains a challenge.

10) *"Furthermore, as expected, manual registration resulted in small inconsistencies across repeated registration compared to an automatic registration." Do you mean that the inconsistencies were smaller for manual registration than for automatic registration? Or that automatic registration resulted in smaller inconsistencies than manual registration? Please rephrase to improve clarity.*

In manual registrations, there are small inconsistencies, but in automatic registration, there are no inconsistencies because the chosen approach is deterministic.

We rephrased the text to make this clearer:

Furthermore, as expected, repeated registration of the same images resulted in small inconsistencies using manual registration, contrary to the automatic (deterministic) registration.

11) "cell level MSI intensity differences between manual and Euler registration, log₂-fold change of example MSI channel of manual vs. Euler registration with the cell area on the x-axis. Boxes show the percentage of cells with a log₂-fold change higher than log₂(1.5), lower than log₂(1.5) or in between." What do the dashed lines represent in this figure? Can you indicate this in the caption? It seems they roughly correspond to log₂(0.5) and -log₂(0.5): did you mean these values in the caption instead of log₂(1.5) and -log₂(1.5)?

Indeed, the dashed lines indeed represent the -log₂(0.5) and log₂(0.5) thresholds used for calculating the percentages. We have fixed the caption for Figure 3.

12) "Quantitatively, the observed MSI-pixel-level associations between m/z 728.52 and Mel624 are shown in Figure 4c". Is it possible to quantify this association with a number in the same image? Maybe a correlation coefficient? Would it be the same scaled coefficient of association shown in Figure 4d?

We added the Pearson correlation coefficient and the slope (of fitting a linear regression) to the scatterplot in Figure 4c. The slope in Figure 4c is closely related to the scaled coefficient depicted in Figure 4d. The main difference is that a model that accounts for spatial autocorrelation (simultaneous autoregressive regression) is used and that the resulting slope is linearly scaled for easier visualization.

We further clarified this in the manuscript:

c) Intensity of lipid vs. cell type marker intensities, same order as in b). The slope of fitting a linear model as well as the pearson correlation coefficient are shown additionally. d) Estimates of association for cell line Mel624 and AC16 with various lipids expected to be higher expressed in Mel624. Results for both the homogeneous and the mixture samples are shown. e) same as d) but expected higher expression in AC16. Estimates and error bars (standard deviation) for d) and e) are linearly scaled between -1 and 1 for simpler visualization.

13) *Reanalysis Nunes et al. data - I believe that this paper didn't publish all the data you ideally need for your workflow (e.g. PostMSI ablation marks). Do you think this affected the performance of MIMIC (or probably cannot even be called MIMIC since not all required data were available)? If so, did it affect it significantly? Do you think that the performance of MIMIC on this data could be improved if all the ideal data were available?"*

Indeed, because of the missing microscopy slide scans, MIMIC co-registration could not be used. Instead registration using intensities was done. While the co-registration quality could potentially be increased if MIMIC co-registration could be used (especially for some of the images), it is hard to make claims about increased performance without available registration metrics.

We added a sentence commenting on this in the discussion:

However, missing microscopy images meant an alternative co-registration based on MSI and IMC intensities needed to be employed and direct registration errors could not be calculated.

14) Results - "While the estimates of association do not seem to depend on the abundance of a cell type (top annotation and left heatmap in Figure 6f), the $-\log_{10}$ FDR corrected p-values are, as expected, on average negatively correlated with the proportion of missing m/z values or in other words, positively correlated with number of observations (Supplementary Figure S25". When you write '(top annotation and left heatmap in Figure 6f)' do you mean 'top and left barplot in Figure 6f'?

Indeed, it should be read that the top boxplot annotation and the estimates in the heatmap do not seem to be correlated. We corrected this in the manuscript.

Additional changes

During the review process, we received additional feedback regarding the posted preprint. Two minor points regarding clarifications in the method section were raised: What exact functions from SimpleITK were used for co-registration? And how exactly is the score that is optimized during matching of MSI ablation marks to MSI pixels calculated? We thus addressed these points by minor adjustments in the Methods section of the manuscript.